# DeepCritic: Deliberate Critique with Large Language Models

## Abstract

As Large Language Models (LLMs) are rapidly evolving, providing accurate feedback and scalable oversight on their outputs becomes an urgent and critical problem. Leveraging LLMs as critique models to achieve automated supervision is a promising solution. In this work, we focus on studying and enhancing the math critique ability of LLMs. Current LLM critics provide critiques that are too shallow and superficial on each step, leading to low judgment accuracy and struggling to offer sufficient feedback for the LLM generator to correct mistakes. To tackle this issue, we propose a novel and effective two-stage framework to develop LLM critics that are capable of deliberately critiquing on each reasoning step of math solutions. In the first stage, we carefully curate 4.5K long-form critiques as seed data for supervised fine-tuning. Each seed critique consists of deliberate step-wise critiques that includes multi-perspective verifications as well as in-depth critiques of initial critiques for each reasoning step. Then, we perform reinforcement learning on the fine-tuned model with either existing human-labeled data from PRM800K or our automatically annotated data obtained via Monte Carlo sampling-based correctness estimation, to further incentivize its critique ability. Our developed critique model built on Qwen2.5-7B-Instruct not only significantly outperforms existing LLM critics (including the same-sized DeepSeek-R1-Distill models and GPT-4o) on various error identification benchmarks, but also more effectively helps the LLM generator refine erroneous steps through more detailed feedback.

## 1 Introduction

Large Language Models (LLMs) (OpenAI, 2023; Qwen Team, 2024; Liu et al., 2024) have demonstrated superior performance that even surpasses human capabilities across a wide range of tasks (Jimenez et al., 2024; Luo et al., 2024b; OpenAI, 2024). LLMs achieve strong and generalizable performance by training on human-provided knowledge data (Ouyang et al., 2022). This makes their evolution highly dependent on the human supervision (Bai et al., 2022). However, as LLMs become increasingly intelligent, providing effective and scalable human supervision will become highly challenging, as collecting human feedback would be too costly and difficult (Saunders et al., 2022).

LLM critics (McAleese et al., 2024), which leverage LLMs as critique models, have recently emerged as a promising approach to enabling scalable oversight (Saunders et al., 2022) on evolving LLMs. LLM critics generate critiques of LLM-generated content, which identify flaws and errors to help the LLM generator refine its outputs, paving the way for the automatic supervision and continuously improvement of LLMs (Madaan et al., 2023; Ke et al., 2024; Gao et al., 2024a; Tang et al., 2025).

The effectiveness and the potential of the utilizing LLM critics heavily depends on the inherent critique capabilities of the LLM critics. However, existing studies have shown that current LLM critics still exhibit limited critique capabilities in complex domains such as mathematical reasoning tasks (Zeng et al., 2023; Zheng et al., 2024a), making it difficult to provide accurate and reliable feedback. By analyzing the critiques generated by existing LLM critics on math problems and solutions, we find that their generated critiques are often overly superficial and lack critical thinking, as shown in the examples in Figure 1. In particular, their critiques merely follow and echo the same reasoning process of the original steps, re-iterating rather than challenging them with their own critical reasoning, leading to a premature judgment result. This behavior can lead to two problems: (1) The critiques of reasoning steps lack careful deliberation, leading to low accuracy in the judgment

> **Problem and Solution**
>
> **Problem**: The sum of four consecutive positive even integers is a perfect square. What is the least possible sum?
> **Solution**: **Step1**: Let the four consecutive positive even integers be $(x, x+2, x+4,$ and $x+6)$. Their sum is $x + (x+2) + (x+4) + (x+6) = 4x + 12$.
> **Step2**: We want this sum to be a perfect square, so we want $4x + 12 = y^2$ for some positive integer $y$. This implies that $4x = y^2 - 12 = (y-2)(y+2) - 8$, **so y−2 and y+2 must both be divisible by 4**. **Step3**: ⋯

> **Qwen2.5-7B-Instruct**
>
> Let's review each step of the solution: **Step 1**: ⋯ **Step 2**: We want this sum to be a perfect square, so want $4x + 12 = y^2$ for some positive integer $y$. This implies that $4x = y^2 - 12 = (y-2)(y+2) - 8$, so $y-2$ and $y+2$ must both be divisible by 4. Critique: This step is correct. The equation $4x + 12 = y^2$ is correctly derived, and the reasoning about $y-2$ and $y+2$ being divisible by 4 is correct. **Step 3**: ⋯ **Final Answer**: ⌐-1⌐ ❌

> **DeepCritic-7B-RL (Ours)**
>
> **Critique of Step 1**: ⋯ **Critique of Step 2**: Let's carefully examine Step 2, where the solution states that $4x + 12 = y^2$ implies $4x = y^2 - 12$. This is a correct algebraic manipulation. The solution then claims that $y-2$ and $y+2$ must both be divisible by 4. Let's analyze this claim more closely. Starting from the equation: $4x + 12 = y^2$. we can rearrange it to: $4x = y^2 - 12$. **This is correct. However, the solution then states that y-2 and y+2 must both be divisible by 4. Let's break this down**: If $y^2 = 4x + 12$, then $y^2 \equiv 0$ mod 4. This means that $y^2$ is divisible by 4. Therefore, $y$ must be divisible by 2 (since the square of an odd number is congruent to 1 modulo 4, and the square of an even number is congruent to 0 modulo 4). So, $y = 2k$ for some integer $k$. Substituting $y = 2k$ into the equation, we get: $4x + 12 = (2k)^2$, $4x + 12 = 4k^2$, $4x = 4k^2 - 12$, $x = k^2 - 3$. This shows that $x$ is an integer if $k^2 - 3$ is an integer, which it is. Therefore, $y$ being even is a necessary condition, **but the solution's claim that y-2 and y+2 must both be divisible by 4 is too strong. To further validate this**, let's consider the correct conditions: $y$ must be even, so $y = 2k$. Substituting $y = 2k$ into the equation $4x + 12 = y^2$ gives $4x + 12 = 4k^2$, which simplifies to $x = k^2 - 3$. This shows that $x$ is an integer as long as $k^2 \geq 3$. Therefore, the correct condition is that $y$ must be even, not that $y-2$ and $y+2$ must both be divisible by 4. **Reflecting on this, the solution's claim is incorrect.** The correct condition is that $y$ must be even, which ensures that $x$ is an integer. Judgement: The correctness of Step 2 is: ⌐-1⌐. **Answer**: ⌐2⌐. ✅

Figure 1: Comparison of critiques generated by current LLM critics and our developed critic. The red highlights in the solution steps represent the erroneous part. The critiques of current LLM critics (e.g., Qwen2.5-7B-Instruct) are overly superficial, primarily consisting of declarative statements rather than in-depth analysis or critical evaluation. In contrast, our critic can generate a deliberate reasoning process before making a judgment, incorporating **iterative evaluation**, **multi-perspective verification**, and **meta-critiquing**. More examples are in Appendix N.

results. (2) The critiques lack informativeness, offering limited guidance for the LLM generator to perform targeted refinements.

In this work, we aim to address the aforementioned limitations of shallow critiques generated by the LLM critics, particularly in the domain of mathematical reasoning.[1] Specifically, we propose the `DeepCritic` framework, which enhances the critique capabilities of LLMs through a two-stage training pipeline. In the first stage, to enable LLMs to acquire a preliminary capability for generating fine-grained critiques, we first curate a dataset of 4.5K long-form critiques by iteratively prompting Qwen2.5-72B-Instruct (Qwen Team, 2024) to critique on a small subset of labeled solutions in PRM800K (Lightman et al., 2023). Each above critique includes step-wise critiques of all reasoning steps if the solution is correct, or up to the first erroneous step otherwise. When constructing each step-wise critique, we first generate an initial and preliminary critique of the specified reasoning step. Then, in order to enable our critic to conduct critiques more critically and from more diverse perspectives, we further generate an in-depth critique of the initial critique. The in-depth critique is supposed to validate the step from alternative perspectives and critically examine the initial critique itself. Finally, we merge the initial and in-depth critique into one deliberate critique for the specified step. By supervised fine-tuning on the curated and filtered critique data, we obtain an initial critique model that is capable of performing multi-perspective evaluation and meta-critiquing through reflection on and correction of its prior erroneous judgments. Then, in the second stage, we perform reinforcement learning (RL) to the SFT model to further boost its deep critique ability. We perform RL under two different settings based on the source of RL data: (1) When human-labeled data is available, such as PRM800K, we directly use it for RL; (2) Otherwise, we automatically generate annotated RL data through Monte Carlo sampling-based correctness estimation on each reasoning step (Wang et al., 2024) and then perform RL.

Experimental results show that our developed deep critic significantly outperforms existing process reward models (PRMs) and LLM critics (including the advanced DeepSeek-R1-Distilled reasoning models (DeepSeek, 2025) and GPT-4o (OpenAI, 2023)) on various error identification benchmarks, demonstrating the effectiveness of our pipeline in enabling LLM critics to provide more accurate supervision. Furthermore, we demonstrate promising test-time scaling properties for both our deep critic and the LLM generator within our framework: (1) the judgment accuracy of the critic consistently improves with increased test-time sampling effort; and (2) the performance of the LLM

---

[1]Our framework can also be effectively applied to subjective domains, as demonstrated in Appendix M.

generator is effectively enhanced either by employing our deep critic as a verifier in majority voting or through refinement guided by the critic's more informative feedback.

## 2 RELATED WORK

**Critique Ability of LLMs** In the current era of rapidly evolving LLMs, leveraging and improving the critique ability of LLMs to facilitate effective scalable oversight (Bowman et al., 2022; Saunders et al., 2022; Lan et al., 2024a) and superalignment (Burns et al., 2024; Yang et al., 2024b) has become increasingly important. Regarding the utilization of the critique ability of LLMs, LLM-as-a-Judge (Gu et al., 2024) and LLM-as-a-Critic (McAleese et al., 2024) serve as promising directions for facilitating automatic supervision of LLM generations (Zheng et al., 2023), enabling the self-evolution of LLMs (Yuan et al.; Wu et al., 2024a), and refining LLM outputs at test time (Madaan et al., 2023). Benchmarking the critique ability of LLMs (Luo et al., 2023; Lan et al., 2024b; Lin et al., 2024) paves the way to better understanding the potential and limitations of current LLMs on critique tasks. Finally, a series of studies aim to create more powerful critique models on mathematical reasoning (Xiong et al., 2024a; Zheng et al., 2024b; Gao et al., 2024a; Xi et al., 2024; Tang et al., 2025), code generation (McAleese et al., 2024; Xie et al., 2025) or other open-domain tasks (Ke et al., 2024; Sun et al., 2024). Compared with previous approaches (Xie et al., 2025; Tang et al., 2025; Wang et al., 2025) that restricts the trained model to producing only direct verification–style critiques, the key distinction and advantage of our method lie in our innovative iterative critique generation process, which injects multi-perspective evaluation and meta-critiquing behaviors into the seed critique data to encourage LLMs to develop more comprehensive and diverse critique behaviors.

**Reasoning Ability of LLMs** The reasoning abilities of LLMs has long been a topic of widespread interest in the community. Previous studies explore LLM reasoning in various domains, such as code (Roziere et al., 2023; Hui et al., 2024), math Liu et al. (2024); Yang et al. (2024a), commonsense knowledge (Rein et al., 2023; Geva et al., 2021), etc. This work mainly focuses on the math reasoning domain. The studies in this line can be divided into several categories: (1) Designing diverse and challenging math reasoning benchmarks to probe the boundaries of existing LLMs' reasoning abilities (Cobbe et al., 2021; Hendrycks et al., 2021; Gao et al., 2024b). (2) Enhancing the math reasoning abilities of LLMs in the training time either by collecting high-quality math datasets for fine-tuning (Yu et al., 2024; Li et al., 2024), or by proposing advanced optimization algorithms (Lai et al., 2024; Cui et al., 2025). (3) Improving reasoning accuracy by increasing the test-time compute either by performing search-based sampling (Wang et al., 2023; Wu et al., 2024b) with process reward models (PRMs) (Lightman et al., 2023; Wang et al., 2024), or by extending the Chain of Thought (CoT) length (OpenAI, 2024; DeepSeek, 2025; Yang et al., 2025b). This work, taking a pioneer step to have the critique model provide judgments after detailed and deliberate reasoning, can also improve LLMs' math reasoning by providing accurate and detailed feedback on erroneous solutions and assisting LLMs in correcting them.

## 3 METHODOLOGY

### 3.1 PROBLEM FORMULATION

Here, we introduce the preliminaries of our studied critique problem setting. Let $\mathcal{D} = \{(P, S)\}$ denote a dataset comprising pairs of problems $P$ and their corresponding solutions $S$. Each solution $S$ is in a step-by-step format denoted as $S = (s_1, s_2, \cdots, s_n)$, where $s_i$ represents the $i$-th step. Denote $\boldsymbol{\theta}_c$ as an LLM critic whose role is to review each step in $S$ sequentially, identify in which step the first error occurs, and return the step index of that first erroneous step. If all steps are deemed correct, it returns -1 to indicate that the entire solution is correct. The output of the critic can be formulated as

$$(c_1, j_1, \cdots, c_k, j_k, a) \sim \pi_{\boldsymbol{\theta}_c}(\cdot | P, s_1, \cdots, s_n), \tag{1}$$

where $c_i$ represents the CoT critique of step $s_i$, $j_i \in \{1, -1\}$ represents the judgment result (i.e., 1 for correct or -1 for incorrect) indicating the correctness of $s_i$. The final judgment result $a$ is equal to $k$ if $j_k = -1$, indicating the first incorrect step; otherwise, if all $j_i = 1$ for $i \leq k = n$, then $a = -1$.

In this work, we aim to improve the LLM's critique ability and enable it to produce more deliberate and thoughtful critiques $c_i$ before making the judgment result $j_i$, enhancing both the accuracy and quality of its generated critiques.

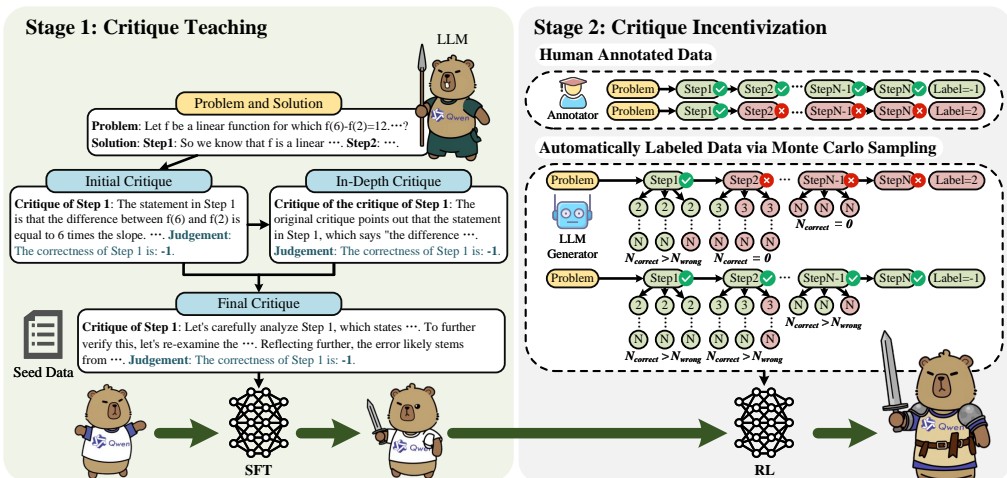

Figure 2: The two-stage pipeline of training our deep critique models. In Stage 1, we first generate an initial step-wise critique for each step in the solution, followed by an in-depth critique of the initial critique. Then, we merge these two critiques into one deliberate critique in the long-CoT form. Finally, we perform SFT on above created critique data to teach the model the format of deliberately critiquing. In Stage 2, we perform RL to the SFT model on either human-annotated data or auto-labeled data via Monte Carlo sampling-based correctness estimation, to further stimulate its critique ability.

### 3.2 DeepCritic: DELIBERATE CRITIQUE MODEL

In this section, we will introduce our two-stage pipeline to create deliberate critique models in detail, including the SFT data generation and training stage in Section 3.2.1, and the RL data curation and optimization stage in Section 3.2.2.

#### 3.2.1 TEACHING LLMs TO DELIBERATELY CRITIQUE

Given that existing LLM critics struggle to produce well-reasoned and in-depth critiques, our first stage aims to **teach** LLMs how to deliberately critique. In this stage, we first leverage Qwen2.5-72B-Instruct (Qwen Team, 2024) to iteratively perform initial and in-depth critiquing,[2] and then merge the two critiques into a single long-form critique to serve as the seed critique. We subsequently perform SFT on the curated seed critique data to teach the target model the format and structure of deliberate critiquing. The brief illustration is displayed in the left part of Figure 2, and the detailed procedure is described below.

**Initial Critique Generation** First, we sample a small set of labeled data from PRM800K Lightman et al. (2023) as the seed task inputs. Each task input contains a problem $P$ and a step-by-step solution $S = (s_1, \cdots, s_n)$, along with the human-labeled label $l_i \in \{1, -1\}$ (for steps that are labeled 0 to indicate no progress rather than incorrectness, we relabel them as 1) indicating the correctness of each reasoning step $s_i$. Thus, we can create the step-by-step critiques on these seed inputs using an LLM $\theta_0$, which is chosen as Qwen2.5-72B-Instruct. However, instead of creating the step-by-step critique of the entire solution directly in a single pass just like Eq. (1), which often leads current LLMs to generate overly brief critiques for each step as mentioned before, we adopt an approach that critiques each step independently. Specifically, in each turn of prompting, we provide the problem and entire solution as inputs, but instruct Qwen2.5-72B-Instruct to critique only one specified step:

$$(c_i^{init}, j_i^{init}) \sim \pi_{\theta_0}(\cdot | P, s_1, \cdots, s_n, s_{target} = s_i), \quad i = 1, \cdots, k, \tag{2}$$

where $s_{target}$ is the additional requirement that specifies the $i$-th step to be critiqued only, $(c_i^{init}, j_i^{init})$ represents the initial CoT critique and judgment result of the specified step, $k$ represents the index of the first step where $l_i = -1$, or $k = n$ when all $l_i$ is 1.

---

[2]Note that our method can be applied in the self-improvement setting, as long as the base model itself has certain initial critique and in-depth critique capabilities. We present preliminary results in Section 4.4. .

**In-Depth Critique Generation** After initial critique generation, we find that many of the initial critiques merely adopt a direct verification approach that directly follows the logic of the original reasoning steps and perform repetitive or similar calculations, resulting in relatively low accuracy when identifying incorrect steps. To enable our critique model to learn to perform critical evaluations, we introduce a second round of in-depth critique generation based on the initial critiques. Specifically, for each reasoning step in the solution, we instruct Qwen2.5-72B-Instruct again to either **assess the reasoning step from a different perspective or using a different evaluation method** than that used in the initial critique, or to **critique the initial critique itself** in order to identify whether there exist flaws in the initial critique that lead to the incorrect judgment about the reasoning step. Therefore, the in-depth critique $c_i^{deep}$ and its judgment result $j_i^{deep}$ are generated as

$$(c_i^{deep}, j_i^{deep}) \sim \pi_{\boldsymbol{\theta}_0}(\cdot | P, s_1, \cdots, s_n, c_i^{init}, j_i^{init}, s_{target} = s_i), \quad i = 1, \cdots, k. \quad (3)$$

This process allows initial critiques that previously led to mismatches between the initial judgment result and the ground truth label (i.e., $j_i^{init} \neq l_i$) to be revised into correct critiques. Then, we only retain the solutions in which the in-depth judgment results of all steps in the solution align with the ground truth labels (i.e., $j_i^{deep} = l_i, \forall i = 1, \cdots, n$) (refer to more discussions in Appendix P), as well as their initial and in-depth critiques. Table 6 shows the proportion of step-level critiques in our final SFT dataset that are successfully corrected through the second-round in-depth critique generation process. We observe that a certain number of step-level critiques (about 1.3K) benefit from the in-depth critique generation process. Incorporating these samples into training equips the model with the capabilities of reflection and self-correction in critiquing (see Figure 10). Additionally, we show the frequency with which Qwen2.5-72B-Instruct adopts multi-perspective evaluation and meta-critiquing during in-depth critique generation in Table 8 for reference. While a few prior studies (Lan et al., 2024a; Sun et al., 2024) also attempt to re-evaluate initial critiques by classifying, filtering, or summarizing them to improve the quality of direct critiques, they do not incorporate meta-critiques into the training samples like our method does. As a result, their trained critique models lack the ability to perform diverse verification and meta-critiquing of prior critiques during inference. Moreover, in Section 4.3, we demonstrate that the introduced in-depth critique generation procedure further enhances model performance compared with training on initial critiques alone.

**Final Critique Synthesis** In the last step of SFT data generation, we use in-context learning with two manually-written examples to instruct Qwen2.5-72B-Instruct to merge the initial and in-depth critiques of each step into a single deliberate critique:

$$(c_i^{final}, j_i^{final}) \sim \pi_{\boldsymbol{\theta}_0}(\cdot | c_i^{init}, j_i^{init}, c_i^{deep}, j_i^{deep}, \{ex_l\}), \quad i = 1, \cdots, k, \quad (4)$$

where $\{ex_l\}$ are in-context learning examples. Finally, we only need to concatenate all step-level critiques to form a complete solution-level critique:

$$C = (c_1^{final}, j_1^{final}, \cdots, c_k^{final}, j_k^{final}, a), \quad a = \begin{cases} k & \text{if } j_k^{final} = -1, \\ -1 & \text{if } j_k^{final} = 1. \end{cases} \quad (5)$$

Such deliberate critiques enable the model to **perform iterative evaluations, multi-perspective verifications, reflection, and meta-critiquing** in the inference stage, thereby improving its judgment accuracy. All generation prompts and hyper-parameters are put in Appendix A and Appendix C.1.

**Supervised Fine-Tuning** In total, we obtain approximately 4.5K seed solution-level critiques, and their label distribution (i.e., the distribution of the step index of the first erroneous step) is shown in Figure 5. We then perform SFT on the base model and obtain an initial critique model $\boldsymbol{\theta}_{SFT}$:

$$\boldsymbol{\theta}_{SFT} = \arg\min_{\boldsymbol{\theta}} \mathbb{E}_{(P,S,C) \sim \mathcal{D}_{SFT}} [-\log P_{\boldsymbol{\theta}}(C | P, S)]. \quad (6)$$

### 3.2.2 Incentivizing LLMs to deliberately critique

Once the seed critique model has acquired a certain level of critique capability, in the second stage, we aim to stimulate and elicit its full potential through continued *incentivization* via RL.

The acquisition of RL data is critical in RL stage. In the following, we explore RL under two different settings based on the sources of data. (1) First, the ideal source for RL should be the high-quality labeled data obtained through expert annotation. Therefore, in the first setting we directly use the existing human-labeled dataset PRM800K (Lightman et al., 2023) for RL. (2) However, in some

cases human annotation may become impractical or even infeasible due to high cost. Thus, in the second setting where human annotation is infeasible, we construct the task data automatically via a Monte Carlo sampling-based correctness estimation method (Wang et al., 2024). Specifically, we sample a portion of GSM8K, MATH and Olympiads problems from NuminaMath-CoT dataset (Li et al., 2024), and leverage Qwen2.5-1.5B/3B/7B-Instruct (Qwen Team, 2024) to generate multiple step-by-step solutions for each problem. Problems where all solutions are either fully correct or fully incorrect are discarded, as such cases are deemed too easy or too challenging. Then, for each incorrect solution, to measure the correctness of a specific step $s_i$, we follow Wang et al. (2024) to truncate the solution after $s_i$, and use an LLM generator (i.e., Qwen2.5-7B-Instruct in this work) to rollout the subsequent reasoning path $N$ times independently. **We define the first erroneous step as the first step from which, along with all subsequent steps, all rollouts generated by the generator are incorrect; while for its all preceding steps, more than half of rollouts reach the correct answers.** If such steps do not exist, we discard those incorrect solutions. For solutions with correct final answers, prior studies (Zheng et al., 2024a; Tang et al., 2025) have pointed out that their intermediate steps can still be incorrect. Therefore, we perform the same Monte Carlo sampling procedure and **assign a label of -1 only to those correct solutions whose all intermediate steps have corresponding rollouts where more than half reach the correct answers**. Our strict data filtering criteria ensures the high quality of the constructed RL data. Furthermore, it is feasible to employ other algorithms, such as binary search (Luo et al., 2024a), to accelerate the data construction process while maintaining annotation quality, thereby enhancing the scalability of our method (refer to Appendix J). We also conduct experiments to investigate the impact of label noise on RL, and the corresponding results and analysis are provided in Appendix K.

The illustration of above data construction procedure is shown in the right part of Figure 2. The detailed data generation settings are put in Appendix C.1. In our experiments, we explore training the SFT model with RL on either 40.7K PRM800K data or 14.3K automatically constructed data. The label distributions of these two data sources are shown in Figure 6 and Figure 7 respectively. We also display the statistics of problem sources in our automatically constructed RL data in Table 7.

## 4 EXPERIMENTS AND ANALYSIS

### 4.1 EXPERIMENTAL SETTINGS

**Base Model** We primarily choose Qwen2.5-7B-Instruct as the base model. We first perform SFT to get our seed critique model **DeepCritic-7B-SFT**. Then, we perform RL on two distinct types of RL data separately, resulting in two variants: **DeepCritic-7B-RL-PRM800K** and **DeepCritic-7B-RL-Numina**. We put the additional results on LLaMA3.1-8B-Instruct (MetaAI, 2024) in Appendix I.

**Benchmarks** We select three widely used error identification benchmarks to systematically evaluate the critique and judgment performance of each model, including the subset of **MR-GSM8K** (Zeng et al., 2023) in which the questions are from original GSM8K (Cobbe et al., 2021) dataset, the Phase-2 test set of **PRM800K** (Lightman et al., 2023), and **ProcessBench** (Zheng et al., 2024a). Each testing example in all datasets contains a problem, a step-by-step solution and a label that either represents the step index of the first erroneous step or is -1 if the solution is entirely correct (see Appendix C.2).

**Baselines** We compare our critique models against two categories of baselines: (1) **Process Reward Models** (PRMs): In this category, we select Math-Shepherd-PRM-7B (Wang et al., 2024), RLHFlow-PRM-8B-Mistral/DeepSeek (Xiong et al., 2024b), Qwen2.5-Math-7B-PRM800K (Zheng et al., 2024a), and Qwen2.5-Math-PRM-7B (Zhang et al., 2025b) for comparison. (2) **LLM Critics**: We prompt the following leading LLMs to serve as critique models: LLaMA3.1-8B/70B-Instruct (MetaAI, 2024), Qwen2.5-7B/72B-Instruct (Qwen Team, 2024), Qwen2.5-Math-7B/72B-Instruct (Yang et al., 2024a), and GPT-4o (Hurst et al., 2024). Also, we include two advanced reasoning LLMs, DeepSeek-R1-Distill-Llama-8B and DeepSeek-R1-Distill-Qwen-7B (DeepSeek, 2025), and use them as reasoning-enhanced critique models for comprehensive comparison.

**Training Settings** In SFT, the learning rate is $1 \times 10^{-5}$, the batch size is $64$, and we fine-tune for 3 epochs. We perform RL based on verl (Sheng et al., 2024) framework, and use Group Relative Policy Optimization (GRPO) (Shao et al., 2024) as RL algorithm. In RL, an accuracy reward of 1.0 is given if the final judgment is correct; otherwise, it is 0.0. We also explored alternative reward designs, such as adding an informativeness-aware reward on top of the accuracy reward, but we do

Table 1: The evaluation results of various PRMs, LLMs that are served as critique models and our critique models on three benchmarks assessing the math critique ability. The underlined value represents the best result for PRMs, while the bold **value** indicates the best result for critique models.

| Model | MR-GSM8K | PRM800K | ProcessBench | | | | Avg. |
| | | | GSM8K | MATH | Olympiad-Bench | Omni-Math | |
|---|---|---|---|---|---|---|---|
| ***Process Reward Models (PRMs)*** | | | | | | | |
| Math-Shepherd-PRM-7B | 61.8 | 21.7 | 48.2 | 27.1 | 20.5 | 16.3 | 32.6 |
| RLHFlow-PRM-8B-Mistral | 66.6 | 25.2 | 50.9 | 32.0 | 13.8 | 15.7 | 34.0 |
| RLHFlow-PRM-8B-DeepSeek | 44.8 | 18.5 | 32.3 | 34.2 | 16.0 | 18.3 | 27.4 |
| Qwen2.5-Math-7B-PRM800K | 70.8 | 55.6 | 70.5 | 64.7 | 50.0 | 42.7 | 59.7 |
| Qwen2.5-Math-PRM-7B | 81.2 | 63.2 | 83.4 | 77.7 | 66.7 | 65.2 | 72.9 |
| *Large Language Models, served as* **Critique Models** | | | | | | | |
| LLaMA3.1-8B-Instruct | 31.6 | 16.0 | 23.8 | 18.9 | 18.3 | 17.2 | 21.0 |
| Qwen2.5-7B-Instruct | 48.1 | 25.6 | 42.9 | 36.6 | 25.5 | 25.9 | 34.1 |
| Qwen2.5-Math-7B-Instruct | 35.6 | 19.4 | 23.1 | 22.0 | 9.2 | 10.4 | 20.0 |
| DeepSeek-R1-Distill-Llama-8B | 69.4 | 55.7 | 65.0 | 62.7 | 58.4 | 51.7 | 60.5 |
| DeepSeek-R1-Distill-Qwen-7B | 77.9 | 57.4 | 71.9 | 69.9 | 56.4 | 46.8 | 63.4 |
| LLaMA3.1-70B-Instruct | 72.4 | 34.1 | 72.5 | 47.6 | 41.0 | 36.8 | 50.7 |
| Qwen2.5-72B-Instruct | 72.6 | 45.3 | 72.2 | 52.4 | 41.9 | 43.1 | 54.6 |
| Qwen2.5-Math-72B-Instruct | 73.6 | 41.0 | 68.6 | 48.5 | 28.6 | 27.3 | 47.9 |
| GPT-4o | 69.7 | 45.9 | 72.1 | 57.3 | 50.5 | 53.4 | 58.2 |
| ***Ablated Critique Models*** | | | | | | | |
| DirectDistill-7B-SFT | 63.9 | 44.9 | 55.7 | 52.1 | 39.1 | 43.1 | 49.8 |
| InitialCritic-7B-SFT | 68.3 | 47.7 | 55.2 | 56.7 | 42.9 | 43.7 | 52.4 |
| ***Our Critique Models*** | | | | | | | |
| DeepCritic-7B-SFT | 67.1 | 48.0 | 59.2 | 61.2 | 46.0 | 43.0 | 54.1 |
| DeepCritic-7B-RL-Numina | 78.6 | 57.1 | 75.2 | 70.0 | 54.3 | 51.2 | 64.4 |
| DeepCritic-7B-RL-PRM800K | **79.1** | **62.7** | **80.0** | **73.2** | **62.9** | **56.7** | **69.1** |

not observe performance gains. The detailed results and discussions are in Appendix L. The detailed training settings are in Appendix C.3.

**Evaluation Settings** In our main evaluation, we use consistent sampling settings across all critique models, with `temperature` set to 0.6, `top_p` to 0.9, and `max_generation_length` to 32K. We only sample once for each task input, while we explore the performance of majority voting in Section 5.1. The evaluation prompt is in Appendix C.4, which is adapted from the official prompt used in ProcessBench (Zheng et al., 2024a). The evaluation metric is the **F1 score** (Zheng et al., 2024a), which is the harmonic mean of the judgment accuracy on the step index of first erroneous step in incorrect solutions and the judgment accuracy on correct solutions. Details are in Appendix C.4.

## 4.2 ERROR IDENTIFICATION RESULTS

The overall results on all benchmarks are displayed in Table 1. We put the detailed results of separate accuracy on both incorrect and correct solutions in Appendix O.

First, we can observe that existing instruct models, especially those of small sizes, exhibit very limited critique capabilities, as reflected in their poor judgment performance. As the model size increases, the corresponding critique capability also increases. Second, improvements in the model's reasoning ability have a positive impact on its critique capability. This is reflected in that the strong math reasoning abilities of the DeepSeek-R1-Distill models can be transferred to the math critique task and bring substantial performance gains. However, we find that DeepSeek-R1-Distill models do not truly learn to critique, as they often rely on directly solving the problems to inform their judgments. This may limit their critique performance on harder problems they struggle to directly solve (e.g., relatively low F1 scores on Omni-Math). Third, **our seed critique model DeepCritic-7B-SFT, trained on 4.5K carefully curated deliberate critique data, achieves a 20-point F1 score improvement (34.1→54.1) over the corresponding base model Qwen2.5-7B-Instruct**. Its overall performance is even comparable to that of Qwen2.5-72B-Instruct. This demonstrates the high quality

of our constructed seed critique data and validates our motivation that teaching LLMs to perform deliberate critiquing can indeed lead to significant performance improvements. In the following Section 4.3, we demonstrate that **such great improvement does not come from naive distillation from Qwen2.5-72B-Instruct, but comes from our carefully designed framework.**

Regarding the RL performance, we can see that **RL with only 14.3K automatically constructed data (i.e., DeepCritic-7B-RL-Numina) can effectively boost the model's critique performance from 54.1 to 64.4**. This validates the potential of automatically constructing supervision data and paves the promising way for the automated scalable oversight. Furthermore, when trained with larger scale RL data with higher quality, **the resulted model DeepCritic-7B-RL-PRM800K significantly outperforms all existing critique models, including GPT-4o and two same-sized DeepSeek-R1-Distill models**. While Qwen2.5-Math-PRM-7B achieves stronger performance than our model, it is important to note that it is trained on a much larger in-house SFT dataset. Furthermore, Qwen2.5-Math-PRM-7B is limited in its ability to provide actionable feedback for solution refinement, as discussed in Section 5.2.2. All in all, the above experimental results demonstrate that our proposed two-stage training paradigm provides new insights and is highly effective in enhancing the critique and verification capabilities of LLMs. We conduct detailed case studies in Appendix N.

### 4.3 ABLATION STUDIES ON THE ITERATIVE SEED CRITIQUE GENERATION PIPELINE

We conduct additional experiments to demonstrate the effectiveness of each stage in the iterative seed critique generation pipeline. We compare DeepCritic-7B-SFT with two baselines. For the first baseline, we use Qwen2.5-72B-Instruct to directly generate a solution-level critique for each candidate solution. Through rejection sampling, we construct an SFT dataset of 4.5K samples to assess the effect of direct distillation. Also, we construct another SFT dataset of 4.5K samples containing only correct initial critiques to evaluate the effectiveness of introducing in-depth critiques. We make sure that both these SFT datasets are comparable in size and label distribution to the SFT dataset used in our main experiments. Then, through SFT, we obtain two variants: **DirectDistill-7B-SFT** and **InitialCritic-7B-SFT**. The comparison results are in Table 1. The conclusions are, (1) **direct distillation from Qwen2.5-72B-Instruct brings limited improvement**, and separated step-level initial critique generation brings more improvement; (2) **incorporating in-depth critiques can further improve the performance of the critique model**, as it encourages the model to reason deeper.

We further display the statistics of self-correction behaviors of our models present in the inference time in Figure 3. Specifically, we randomly sample 10 correct and 10 incorrect solutions from each subset of ProcessBench (~100 reasoning steps for each set), together with the corresponding critiques generated by DeepCritic-7B-SFT and DeepCritic-7B-RL-PRM800K. We then prompt GPT-4o to determine, for each step-level critique, whether self-correction behavior is present. Finally, for each model and each subset, we calculate the proportion of step-level critiques that contained self-correction behaviors. As we can see, **both critique models indeed exhibit certain deliberate reasoning behaviors during inference**. Compared with the SFT model, the RL model exhibits a lower frequency of self-correction. A plausible explanation is that, as the model's critique ability improves through RL training, its initial critiques become more accurate, thereby reducing the frequency for subsequent self-correction.

### 4.4 RESULTS OF SELF-IMPROVEMENT

To demonstrate the scalability of our data curation pipeline, we further show that it does not necessarily rely on a stronger teacher model and can also be used for model's self-improvement. To validate this, we conduct experiments by leveraging Qwen2.5-7B-Instruct itself for curating its own seed critique data by following the pipeline introduced above. Then, we perform SFT and RL (with Numina-14K) on Qwen2.5-7B-Instruct under the same settings as in the main experiments. The results in Table 2 demonstrate **the effectiveness and promise of our method for LLM self-improvement.**

## 5 TEST-TIME SCALING RESULTS

In this section, we explore the test-time scaling properties within the critique framework, from the perspectives of both critics and generators. In the following experiments, we choose DeepCritic-7B-RL-PRM800K as the target model, and refer to it as **DeepCritic-7B-RL** for brevity. We take the

Table 2: The results of self-improvement experiments on Qwen2.5-7B-Instruct. The superscript Self indicates models that are self-improved without relying on data produced by stronger teachers.

| Model | MR-GSM8K | PRM800K | ProcessBench | | | | Avg. |
|---|---|---|---|---|---|---|---|
| | | | GSM8K | MATH | Olympiad-Bench | Omni-Math | |
| Qwen2.5-7B-Instruct | 48.1 | 25.6 | 42.9 | 36.6 | 25.5 | 25.9 | 34.1 |
| DeepCritic-7B-SFT[Self] | 51.2 | 33.3 | 44.4 | 41.1 | 28.9 | 30.1 | 38.2 |
| DeepCritic-7B-RL-Numina[Self] | 78.6 | 57.1 | 75.2 | 70.0 | 54.3 | 51.2 | 64.4 |

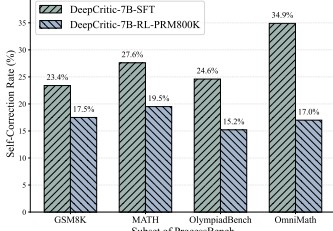

Figure 3: The statistics of self-correction behaviors of our models in inference time.

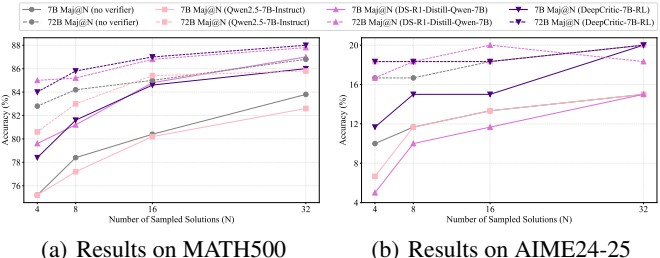

(a) Results on MATH500      (b) Results on AIME24-25

Figure 4: Verified majority voting results of Qwen2.5-7B/72B-Instruct on MATH500 and AIME24-25.

base model Qwen2.5-7B-Instruct along with the strongest baseline DeepSeek-R1-Distill-Qwen-7B (abbreviated to DS-R1-Distill-Qwen-7B) for comparison.

## 5.1 TEST-TIME SCALING RESULTS OF CRITICS

Here, we investigate the effectiveness of the majority voting practice (Wang et al., 2023) in enhancing the critique performance. For each critique model, the final judgment on each input is the majority voting result over eight samplings (Maj@8). We leave the comparison results in Figure 8 in Appendix G, while we highlight the conclusions here: (1) The majority voting practice improves performance across all models. (2) **The Maj@8 results of our critique model outperforms baselines on all benchmarks**, demonstrating the good test-time scaling property of our critique model.

## 5.2 TEST-TIME SCALING RESULTS OF GENERATORS

Critics can enhance LLM generators by scaling their test-time compute. Similar to PRMs, they can serve as verifiers to filter out incorrect responses, thereby improving majority voting accuracy. Additionally, critics can provide informative feedback to help generators to refine potentially erroneous responses. In the following, we delve into these two aspects separately using Qwen2.5-7B-Instruct and Qwen2.5-72B-Instruct as generators, evaluated on MATH500 and AIME2024–2025. For answer correctness verification, we adopt the rule-based verifier `Math-Verify`.

### 5.2.1 RESULTS OF VERIFIED MAJORITY VOTING

We record the majority voting results under different numbers of sampled solutions from the generators, filtered taking the critique model as the verifier. If all candidate solutions are identified as incorrect, we perform majority voting over the original solutions. In this and following refinement experiments, we adjust generators' system prompt to ensure the generators produce responses in a required step-by-step format for critique, which is displayed in Appendix F. We display the results in Figure 4. We observe that when the critique model performs poorly (e.g., Qwen2.5-7B-Instruct), using it as the verifier in majority voting can be counterproductive. In contrast, our critique model can yield greater improvements to generators' majority voting performance in most sampling settings.

Table 3: Results of critique-based refinement. "w→c" denotes the proportion of cases where a wrong solution becomes correct after judgment and refinement, while "c→w" indicates the opposite. "Acc." represents the average accuracy on all testing samples. "*" denotes that the refinement results are biased due to DS-R1-Distill-Qwen-7B directly producing the correct answers during critique.

| Critique Model | Qwen2.5-7B-Instruct | | | | | | Qwen2.5-72B-Instruct | | | | | |
| | MATH500 | | | AIME24-25 | | | MATH500 | | | AIME24-25 | | |
| | w→c | c→w | Acc. | w→c | c→w | Acc. | w→c | c→w | Acc. | w→c | c→w | Acc. |
| *before refinement* | | | | | | | | | | | | |
| | — | — | 74.00 | — | — | 6.67 | — | — | 77.00 | — | — | 11.67 |
| *after refinement* | | | | | | | | | | | | |
| Qwen2.5-7B-Instruct | 0.80 | 2.60 | 72.20 | 1.67 | 0.00 | 8.33 | 1.60 | 2.40 | 76.20 | 1.67 | 0.00 | 13.33 |
| Qwen2.5-Math-7B-PRM | 1.20 | 0.40 | 74.80 | 3.33 | 0.00 | 10.00 | 3.80 | 1.00 | 79.80 | 3.33 | 1.67 | 13.33 |
| DeepCritic-7B-RL | 5.40 | 2.00 | **77.40** | 8.33 | 1.67 | **13.33** | 6.00 | 2.40 | **80.60** | 5.00 | 1.67 | **15.00** |
| *after refinement (answer leakage)* | | | | | | | | | | | | |
| DS-R1-Distill-Qwen-7B* | 7.20 | 1.20 | 80.00 | 8.33 | 0.00 | 15.00 | 7.40 | 1.00 | 83.40 | 3.33 | 0.00 | 15.00 |

### 5.2.2 RESULTS OF CRITIQUE-BASED REFINEMENT

In this setting, we first prompt generators to produce step-by-step solutions for each problem. Then, we leverage each critic to critique the solutions and prompt the generators to revise those deemed incorrect, based on the critic's feedback. We use greedy decoding for the generators for determinism. In experiments, we observe that DS-R1-Distill-Qwen-7B frequently continues critiquing until the end of the solution and produces the correct answer, even though we explicitly instruct the model in the system prompt to stop after identifying the first incorrect step (i.e., poor instruction-following ability). This issue can cause the refinement results to be biased and greatly influenced by DS-R1-Distill-Qwen-7B's own problem-solving capability. Therefore, we present its results independently for reference. The refinement results are shown in Table 3. We can see that our critique model can effectively assist the generators in correcting errors by providing more detailed feedback, leading to improved performance of the generators. Notably, our 7B critique model is also capable of supervising and correcting the outputs of a 72B generator, demonstrating a potential of weak-to-strong supervision Burns et al. (2024).[3] Also, notice that although Qwen2.5-Math-7B-PRM performs better on error identification benchmarks, it shows limited effectiveness on helping generator's to perform refinement. This highlights the superior advantage of our critique model over Qwen2.5-Math-7B-PRM in assisting the generator to improve its solutions.

Moreover, DeepCritic can provide effective online feedback to help the policy model correct errors in its generated wrong solutions, enabling it to produce correct refinements especially for problems that it had never solved before during RL, thereby substantially improving training effectiveness. Existing works (Zhang et al., 2025a; Xie et al., 2025) have also demonstrated the benefits of such critique-driven feedback in online RL, and our critique models can provide a strong initialization for the critic components in their frameworks. We believe exploring the potential of incorporating DeepCritic into online RL training can be an interesting and practical future direction.

## 6 CONCLUSION

In this work, we propose an effective pipeline to enhance the math critique ability of LLMs. We first carefully construct 4.5K long-form critiques incorporating multi-perspective verification and meta-critiquing. These serve as seed data for SFT, enabling the target model to acquire an initial ability of deliberately critiquing. We then further enhance the critique capability of the model via RL. The deep critique model we developed demonstrates superior performance across a range of error identification benchmarks, and exhibits promising potential in supervising and improving the reasoning capabilities of LLM generators that are even more capable than itself. We hope our work provides valuable insights to advance future research in deliberate reasoning and scalable oversight.

---

[3]We put the additional results on GPT-4o (Hurst et al., 2024) and GPT-4.1 (OpenAI, 2025) in Appendix H to show the effectiveness of critique-based refinement by DeepCritic-7B-RL on supervising the stronger generators.

## ETHICS STATEMENT

This work aims to investigate and address the issue that current LLM critics produce overly superficial critiques in mathematical reasoning tasks, leading to low error identification accuracy and insufficient feedback for error correction. We propose a novel two-stage training framework to greatly enhance the critique capabilities of LLM critics. We hope our work can provide new insights to future research on automated and scalable oversight, and further enhance the problem-solving capabilities of LLM generators by improving the critique model.

## REPRODUCIBILITY STATEMENT

First, we provide the code and data in the supplementary material to ensure reproducibility. Then, we provide the detailed data construction procedure in Section 3.2 and display the used prompts in Appendix A. We provide detailed experimental settings and hyper-parameters in Section 4.1 and Appendix C. We display the data statistics in Appendix D and Appendix E.

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

# A    PROMPT TEMPLATES FOR CRITIQUE DATA GENERATION

## Prompt Template for Initial Critique Generation

You are a math expert and are tasked with evaluating the solution path for a mathematical problem.
The solution is presented as a step-by-step chain of thought.
Each step is separated with the index indicator "Step i:", with i indexed starting from 1.
You are required to only critique the specific step carefully and comprehensively.
You need to thoroughly consider the logical consistency of the specified step with the problem statement and previous steps, ensuring each step aligns with the overall and correct objective.
You should consider the cases where the steps are merely irrelevant transitions as correct if there is no critical information missing.
For steps involving numerical calculations, carefully verify the accuracy of the calculations to ensure all results are correct.
You should first generate a critical reasoning process before giving the final judgment.

##Format for Evaluation##
For each specified step in the solution path, perform your evaluation by following the below format:
**Critique of Step <current_step>**: First generate a detailed reasoning thought to evaluate the step.
**Judgement**: Based on the above critique, give your final judgement in the form of "#### The correctness of Step <current_step>is: \boxed{1 | -1}", where 1 represents correct and -1 represents incorrect. The judgement result should be either 1 or -1.

<Problem>{problem} </Problem>

<Solution Path>{solution} </Solution Path>

Now, please critique Step {step_index} in the above solution path.

## Prompt Template for In-Depth Critique Generation

You are a math expert and are tasked with evaluating the critique for a specific step in a solution to a mathematical problem.
You will be given the problem, the solution path, and the critique for a specified step in the solution path.
You need to critique the critique for the specified step and provide your judgement on whether the critique is correct or incorrect, and then determine the final correctness of the specified step.
You need to think about how you would approach evaluating the step if you were asked to do so, without referring to the original critique.
You can either re-evaluate the specified step using different valid approaches or from different perspectives than the original critique to see if different methods can reach the same conclusion; or alternatively, you can critique the original critique itself to check if it is correct and whether it is fair and reasonable.
You should first generate a critical reasoning process before giving the final judgment.

##Format for Evaluation## Perform your evaluation to the critique by following the below format:
**Critique of the critique of Step <current_step>**: First generate a detailed critique either by re-evaluating the specified step with different ways or by directly evaluating the original critique of the step.
**Judgement**: Based on the results of original critique and critique's critique, give your final judgement on the correctness of the specified step in the form of "#### The correctness of Step <current_step>is: \boxed{1 | -1}", where 1 represents correct and -1 represents incorrect. The judgment result should be either 1 or -1.

<Problem>{problem} </Problem>

<Solution Path>{solution} </Solution Path>

<Original Critique>{original_critique} </Original Critique>

Now, please critique the original critique of the Step {step_index} and give your final judgement on the correctness of Step {step_index}.

## Prompt Template for Final Critique Synthesis

You are a math expert and a good math critic.
You will be provided with an initial critique and a critique of the initial critique.
Your task is to merge the two critiques into a single, deliberate critique.
You should merge the two critiques as if they were generated in one go, as if the model first generated a critique and then wanted to further verify that step or the critique itself.
You should make the merged critique smooth by adding some transitional, pausing, reflective, thinking words or sentences. Do not use terms like "the original critique" as the merged critique should be considered as generated in one go.

Here are two examples that can serve as references for the tone and format of the merged deliberate critique:

<Merged Deliberate Critique Example 1>{example1}</Merged Deliberate Critique Example 1>

<Merged Deliberate Critique Example 2>{example2}</Merged Deliberate Critique Example 2>

Please follow the above examples to generate the merged deliberate critique for the below sample:

<Original Critique>{original_critique}</Original Critique>

<Critique of the Original Critique>{critique_of_original_critique}</Critique of the Original Critique>

# B    STATEMENT ON THE USE OF LLMS

In this work, we employ LLMs (specifically, Qwen2.5-72B-Instruct) to construct our SFT and RL datasets, thereby reducing human labor costs (refer to Section 3.2). We also use LLMs to polish the writing; however, they are not used for full paper writing.

Table 4: Training hyper-parameters in SFT.

| Hyper-parameter | Value |
|---|---|
| LR | $1 \times 10^{-5}$ |
| LR Scheduler | cosine |
| Batch Size | 64 |
| Epochs | 3 |
| Maximum Sequence Length | 16384 |
| Warmup Ratio | 0.1 |

Table 5: Training hyper-parameters in RL.

| Hyper-parameter | Value |
|---|---|
| Train Batch Size | 128 |
| Micro Batch Size | 128 |
| Rollout $n$ | 16 |
| Maximum Prompt Length | 2048 |
| Maximum Response Length | 8192 |
| Temperature | 1.0 |
| Top $p$ | 1.0 |
| LR | $1 \times 10^{-6}$ |
| Epochs | 2 |
| KL Coefficient | 0.001 |

## C  DETAILED EXPERIMENTAL SETTINGS

### C.1  HYPER-PARAMETERS IN THE DATA GENERATION STAGE

In the original PRM800K dataset, there are some steps labeled with 0, indicating that these steps are not incorrect but do not make any progress. We consider the ground truth label for these steps to be 1. In the initial critique generation stage, for each reasoning step in the given solution, we prompt Qwen2.5-72B-Instruct once with `temperature = 0.7` and `top_p = 0.9`. In the in-depth critique generation stage, based on the problem, solution and the initial critique, we prompt Qwen2.5-72B-Instruct 16 times with `temperature = 1.0` and `top_p = 0.9`. We then randomly select one critique from those whose judgment results $j_i^{deep}$ are consistent with the ground truth labels, and use it as the in-depth critique for the corresponding step. We only retain the solutions in which the in-depth judgment results of all steps in the solution align with the ground truth labels (i.e., $j_i^{deep} = l_i, \forall i = 1, \cdots, n$), as well as their initial and in-depth critiques. Finally, during critique merging, the sampling parameters for Qwen2.5-72B-Instruct are `temperature = 0.7` and `top_p = 0.9`.

### C.2  OVERVIEW OF EVALUATION BENCHMARKS

For MR-GSM8K (Zeng et al., 2023), we only choose the subset in which the questions are from the original GSM8K (Cobbe et al., 2021) dataset, containing 693 correct solutions and 725 incorrect solutions. For PRM800K (Lightman et al., 2023), we adopt the Phase-2 test set that contains 586 correct solutions and 2078 incorrect solutions. The third evaluation benchmark is ProcessBench (Uesato et al., 2022), which includes four subsets in which the questions originates from different sources: GSM8K, MATH (Hendrycks et al., 2021), OlympiadBench (He et al., 2024) and Omni-Math (Gao et al., 2024b). The total number of solutions for these four subsets are 400, 1000, 1000, and 1000, respectively.

### C.3  DETAILED TRAINING SETTINGS

The complete training hyper-parameters in SFT and RL are put in Table 4 and Table 5 respectively. In RL, an accuracy reward of 1.0 is given if the final judgment is correct; otherwise, it is 0.0. During RL, we observe that in very few cases, the policy model generates critiques that are mixed with different languages, which is consistent with the findings in DeepSeek-R1 (DeepSeek, 2025). However, this issue gradually diminishes as training progresses, so we do not introduce a language consistency reward here. The training is conducted on 8 * NVIDIA A800 (80G).

### C.4  DETAILED EVALUATION SETTINGS

In the main evaluations (Table 1), we use consistent sampling settings across all critique models, with `temperature` set to 0.6, `top_p` to 0.9, and `max_generation_length` to 32K during inference. We only sample once for each task input in this setting. We then calculate the judgment accuracy

on the step index of first erroneous step in incorrect solutions and the judgment accuracy on correct solutions. The F1 score is then calculated as the harmonic mean of the above two numbers.

In the experiments of majority voting (Section 5.1), for each selected critique model, we sample 8 responses with `temperature` set to 1.0 and `top_p` to 0.9.

The evaluation prompt is mainly based on that used in Zheng et al. (2024a), and is put in below. We do not directly use the prompt used in Zheng et al. (2024a), because we find that it will make the model tend to directly generate the judgment results without the reasoning process. All evaluations are conducted on 4 * NVIDIA A100 (80G).

---

**Evaluation Prompt**

The following is a math problem and a solution.
The solution is presented as a step-by-step chain of thought.
Each step is separated with the index indicator "Step i:", with i indexed starting from 1.

<Problem>
{problem}
</Problem>

<Solution>
{tagged_response}
</Solution>

Your task is to evaluate the solution and identify the earliest step that contains an error, or confirm that all steps are correct.
Please first review and generate a critique for each step.
After reviewing each step, once you identify an error in the step, stop the critique and return the index of that step as this is the step where the earliest error occurs. Otherwise, continue reviewing the subsequent steps. If all steps are correct, return the index of -1 (which typically denotes "not found").
Finally, put your final answer (i.e., the step index) in \boxed{}.

---

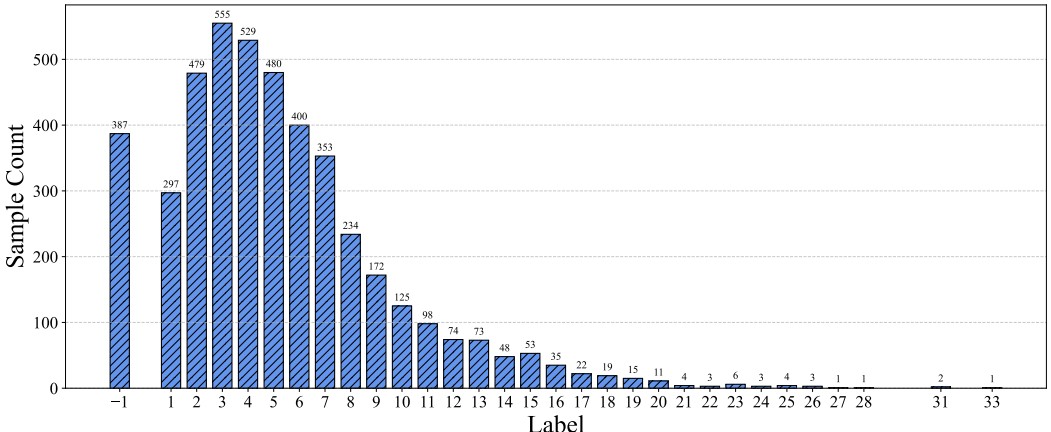

Figure 5: Solution-level label distribution of seed critique data in SFT stage.

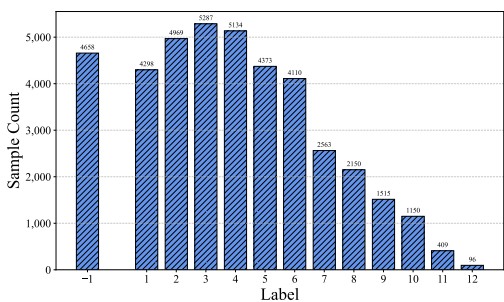

Figure 6: Solution-level label distribution of RL data based on PRM800K.

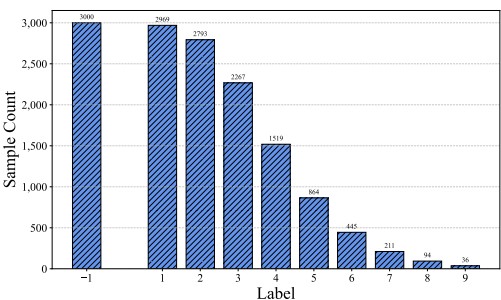

Figure 7: Solution-level label distribution of RL data based on NuminaMath-CoT.

Table 6: Statistics of step-level critiques in our SFT dataset, categorized by the correctness of their corresponding initial critiques.

| Label of Reasoning Step | # Correct Initial Critiques | # Incorrect Initial Critiques |
|---|---|---|
| 1 | 22968 | 738 |
| -1 | 3535 | 565 |

Table 7: The statistics of problem sources in the auto-constructed 14K RL data.

| Source | # Problems | Proportion (%) |
|---|---|---|
| GSM8K | 5483 | 38.4% |
| MATH | 5359 | 37.6% |
| Olympiads | 3424 | 24.0% |
| **Total** | **14266** | **100.0%** |

## D  STATISTICS OF SFT AND RL DATA

We show the solution-level label distribution (i.e., he distribution of the step index of the first erroneous step) of our curated SFT dataset in Figure 5. The label distributions of RL data based on PRM800K and NuminaMath-CoT are show in Figure 6 and Figure 7 respectively. We note that due to limited computational resources, we set the `max_response_length` to 8192 during the RL phase. As a result, **we apply filtering and constraints to the RL data, retaining only task inputs whose solutions contain a number of reasoning steps within a certain range**. This avoids the situation where a large number of rollouts would be truncated due to excessive lengths, which could degrade training performance. However, we believe that with sufficient computational resources, incorporating inputs with more steps and increasing the `max_response_length` during RL can further improve RL performance.

Furthermore, we display the proportion of step-level critiques in our final SFT dataset that are successfully corrected through the second-round in-depth critique generation process in Table 6. We also display the statistics of problem sources in our automatically constructed RL data in Table 7 to demonstrate the diverse distribution of problem difficulty in our curated dataset.

## E  STATISTICS ON THE FREQUENCY OF MULTI-PERSPECTIVE EVALUATION AND META-CRITIQUING ADOPTED BY THE LLM IN-DEPTH CRITIC

We randomly select 70 SFT samples containing a total of 485 final step-level critiques, and count the frequency of multi-perspective evaluation and meta-critiquing adopted by Qwen2.5-72B-Instruct in the in-depth critique generation stage by prompting GPT-4. The results are shown in Table 8.

## F  PROMPT TEMPLATES IN VERIFIED MAJORITY VOTING AND CRITIQUE-BASED REFINEMENT EXPERIMENTS

The prompt templates to create step-by-step solutions and critique-based refinements for generators are put below separately. In the experiments of critique-based refinements, we further add instructions to instruct DeepSeek-R1-Distill-Qwen-7B not to produce the final answer and to stop critiquing after identifying the first incorrect step. However, even with this constraint, we still find that in a certain number of cases, DeepSeek-R1-Distill-Qwen-7B does not follow the instruction and continues critiquing until the end.

---

**Prompt Template for Generating Step-by-Step Solutions**

System: You are a helpful assistant. Please reason step by step, and put your final answer within \boxed{}.
User: {problem}
Assistant: Step 1:

---

Table 8: The frequency of multi-perspective evaluation and meta-critiquing adopted by Qwen2.5-72B-Instruct in the in-depth critique generation stage.

| Only Multi-Perspective Evaluation | Only Meta-critiquing | Both |
|---|---|---|
| 15.1% | 40.0% | 44.9% |

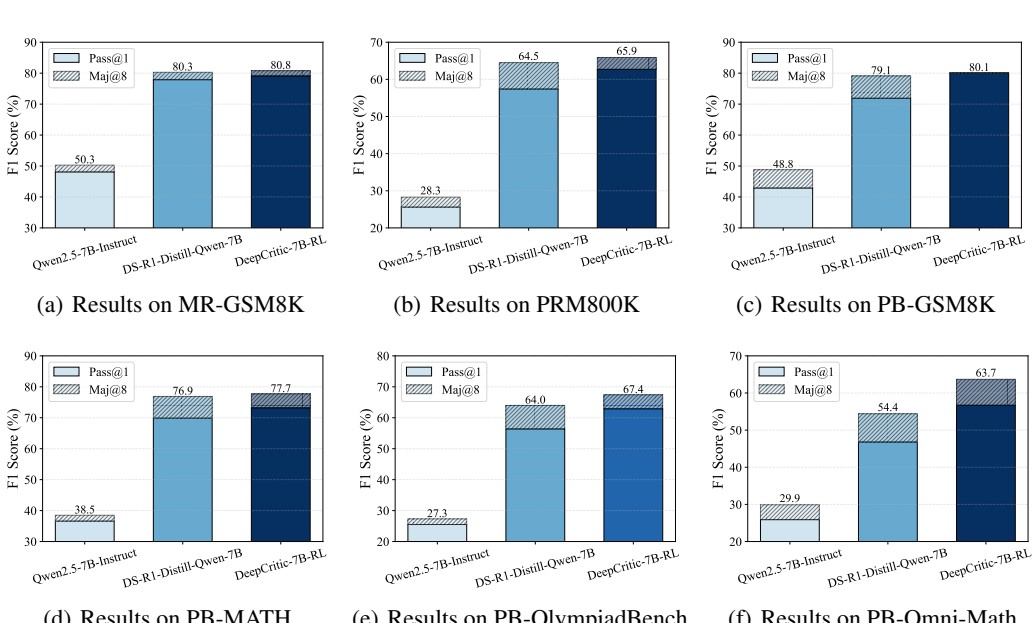

(a) Results on MR-GSM8K
(b) Results on PRM800K
(c) Results on PB-GSM8K

(d) Results on PB-MATH
(e) Results on PB-OlympiadBench
(f) Results on PB-Omni-Math

Figure 8: Majority voting results (Maj@8) of each model across all benchmarks. Pass@1 results are from Table 1. "PB" denotes ProcessBench.

---

**Prompt Template for Critique-Based Refinement**

System: You are a helpful assistant. Please reason step by step, and put your final answer within \boxed{}.
User: {problem}
Assistant: {initial_solution}
User: There might be some problems in your solution. Here is the critique of the above solution:
{critique}
Please carefully refine the solution based on the critique.
Assistant:

---

## G  MAJORITY VOTING RESULTS OF CRITICS

We display the detailed majority voting results of our model DeepCritic-7B-RL and two baselines, Qwen2.5-7B-Instruct and DeepSeek-R1-Distill-Qwen-7B, in Figure 8. As shown, the majority voting practice improves performance across all models. **The Maj@8 results of our critique model outperforms DS-R1-Distill-Qwen-7B in all cases, demonstrating the good test-time scaling property of our critique model.**

## H  CRITIQUE-BASED REFINEMENT RESULTS ON GPT-4O AND GPT-4.1

Here, we further use GPT-4o and GPT-4.1 as the generators, and evaluate the effectiveness of critique-based refinement by DeepCritic-7B-RL (i.e., DeepCritic-7B-RL-PRM800K) on supervising the stronger generators. The results are in Table 9. We can see that **our critique model can also effectively supervise the outputs produced by stronger generators**.

Table 9: Results of critique-based refinement by using GPT-4o and GPT-4.1 as the generators. The results are evaluated on MATH500. "w→c" denotes the proportion of cases where a wrong solution becomes correct after judgment and refinement, while "c→w" indicates the opposite. "Acc." represents the average accuracy on all testing samples.

| Critique Model | GPT-4o | | | GPT-4.1 | | |
|---|---|---|---|---|---|---|
| | w→c | c→w | Acc. | w→c | c→w | Acc. |
| *before refinement* | | | | | | |
| | — | — | 77.40 | — | — | 90.80 |
| *after refinement* | | | | | | |
| Qwen2.5-7B-Instruct | 1.60 | 0.60 | 78.40 | 0.40 | 0.20 | 91.00 |
| DeepCritic-7B-RL | 3.60 | 1.20 | **79.80** | 1.00 | 0.40 | **91.40** |

Table 10: Results on LLaMA3.1 models on ProcessBench.

| Model | ProcessBench | | | | Avg. |
|---|---|---|---|---|---|
| | GSM8K | MATH | Olympiad-Bench | Omni-Math | |
| *8B Models* | | | | | |
| LLaMA3.1-8B-Instruct | 23.8 | 18.9 | 18.3 | 17.2 | 19.6 |
| DeepCritic-LLaMA-8B-SFT | 57.1 | 48.5 | 38.7 | 36.4 | 45.2 |
| DeepCritic-LLaMA-8B-RL-Numina | 72.0 | 59.4 | 44.7 | 42.2 | 54.6 |
| DeepCritic-LLaMA-8B-RL-PRM800K | **74.2** | **62.5** | **46.9** | **44.9** | **57.1** |
| *70B Models* | | | | | |
| LLaMA3.1-70B-Instruct | 72.5 | 47.6 | 41.0 | 36.8 | 49.5 |

## I  ADDITIONAL RESULTS ON LLAMA3.1-8B-INSTRUCT

We also conduct our two-stage training by taking LLaMA3.1-8B-Instruct (MetaAI, 2024) as the base model. The results on ProcessBench are shown in Table 10. The results indicate that our method can also improve the critique performance of LLaMA3.1-8B-Instruct, demonstrating the generalizability of our method.

## J  RESULTS OF USING BINARY SEARCH FOR RL DATA CONSTRUCTION

In the RL data construction of our main experiments, we perform Monte Carlo-based sampling for each reasoning step of every solution, resulting in a time complexity of $\mathcal{O}(N)$ (where $N$ is the number of reasoning steps). Here, we explore using *binary search* (Luo et al., 2024a) to accelerate the auto-annotation process. Binary search works by repeatedly selecting the midpoint of the remaining partial solution, performing rollouts from that point, and determining whether the first incorrect step occurs in the first or second half, reducing the time complexity to $\mathcal{O}(\log N)$. Here, the criterion for determining whether the previous half contains an erroneous step is as follows: if more than half of the rollouts sampled from the middle step are correct (consistent with the criterion used in our main paper for assessing the correctness of a step), then the previous half is considered correct, and the first erroneous step appears in the subsequent half. We then curate 14K data, with a size and label distribution comparable to those used in our main experiments, for RL. The evaluation results are in Table 11. We can observe that **using binary search may slightly degrade performance due to the larger label noise introduced by a weaker filtering criterion, but it significantly improves annotation efficiency, which enhances the scalability of our data construction pipeline.**

## K  THE IMPACT OF LABEL NOISE ON RL

To examine the impact of label noise on RL performance, we conduct additional experiments using Numina-14K, our curated dataset curated under stringent filtering procedures and therefore exhibiting

Table 11: The ablation results of using binary search for auto-annotation on RL data, training on noisy RL data, and introducing informativeness-aware reward during RL.

| Model | MR-GSM8K | PRM800K | ProcessBench | | | | Avg. |
|---|---|---|---|---|---|---|---|
| | | | GSM8K | MATH | Olympiad-Bench | Omni-Math | |
| DeepCritic-7B-SFT | 67.1 | 48.0 | 59.2 | 61.2 | 46.0 | 43.0 | 54.1 |
| DeepCritic-7B-RL-Numina | **78.6** | **57.1** | **75.2** | **70.0** | 54.3 | 51.2 | **64.4** |
| *w/ Binary Search for Auto-Annotation* | 77.0 | 56.0 | 72.1 | 67.5 | **57.0** | **53.8** | 63.9 |
| *w/ 20% Label Noise in Wrong Solutions* | 76.4 | 55.7 | 72.1 | 68.1 | 55.8 | 50.3 | 63.1 |
| *w/ Informativeness-aware Reward* | 76.9 | 51.2 | 67.6 | 62.4 | 49.7 | 46.3 | 59.0 |

reliable label accuracy. Specifically, we randomly corrupt 20% of the samples with wrong final answers in Numina-14K by modifying their labels (i.e., the first erroneous step's step index) to random incorrect values, producing Numina-14K-Noisy. We then perform RL on DeepCritic-7B-SFT using Numina-14K-Noisy, and put the comparison results in Table 11. As we can see, **although the presence of noise in the RL data degrades the final performance slightly, the model is still able to improve through RL even under noisy conditions, demonstrating a certain degree of robustness**. The plausible explanation is that, in GRPO, if the label is corrupted and given the relatively large label space in our task, the rollout within that group are highly likely to all receive a reward of 0. As a result, the entire group does not contribute to the update, thereby avoiding any negative impact on the model (in few cases, if incorrect rollouts happen to produce answers that match the corrupted label, then the training can indeed be affected).

## L   RESULTS OF ALTERNATIVE REWARD DESIGNS

Here, we explore the effectiveness of alternative reward designs in RL. we conduct additional experiments by augmenting the original judgment correctness reward with an additional informativeness-aware reward produced by an LLM judge. Specifically, during RL, we use Qwen3-4B-Instruct-2507 (Yang et al., 2025a) to evaluate the content and the depth of each critique: (1) If a critique exhibits self-correction behavior that leads to the correct final judgment, we add +1 to the original accuracy reward. (2) if no self-correction behavior is present, the additional reward remains 0. (3) If self-correction occurs but leads to an incorrect final judgment, we add –1 to the original accuracy reward. The comparison results are shown in Table 11. We have two observations: (1) **Introducing informativeness-aware reward does not lead to improvements in judging performance.** Accuracy reward is the most relevant and effective signal for improving critique performance. (2) **Introducing an informativeness-aware reward substantially increases the average training response length from 2,000 to 2,800, which may indicate a form of reward hacking**: the policy model produces excessive or uninformative self-reflection in an attempt to obtain higher informativeness-aware rewards, and this ineffective self-reflection directly contributes to the inflated response length.

## M   GENERALIZATION TO SUBJECTIVE DOMAINS

In the main experiments, we validated the effectiveness of our `DeepCritic` framework in enhancing the critique capabilities of LLMs in verifiable domains (e.g., mathematics). Here, we provide additional results of applying our pipeline to a subjective domain, *text summarization*, to demonstrate the generalizability of our method to open domains.

Specifically, we follow the previous study (Stiennon et al., 2020) to first use `TL;DR` dataset (Völske et al., 2017) to fine-tune Qwen2.5-7B-Instruct and get **Qwen2.5-7B-Summary-SFT**. We also train a reward model **Qwen2.5-7B-Summary-RM** on the provided preference data from Stiennon et al. (2020). Then, we leverage Qwen2.5-7B-Summary-SFT to generate summaries on a held-out set and follow the same iterative critique-generation pipeline outlined in the main paper to produce deliberate critiques for each summary. To filter useful critiques for training, we first use Qwen2.5-7B-Summary-SFT to refine summaries based on the corresponding critiques, and then score the initial summaries and their refinements using Qwen2.5-7B-Summary-RM. We filter out the critiques

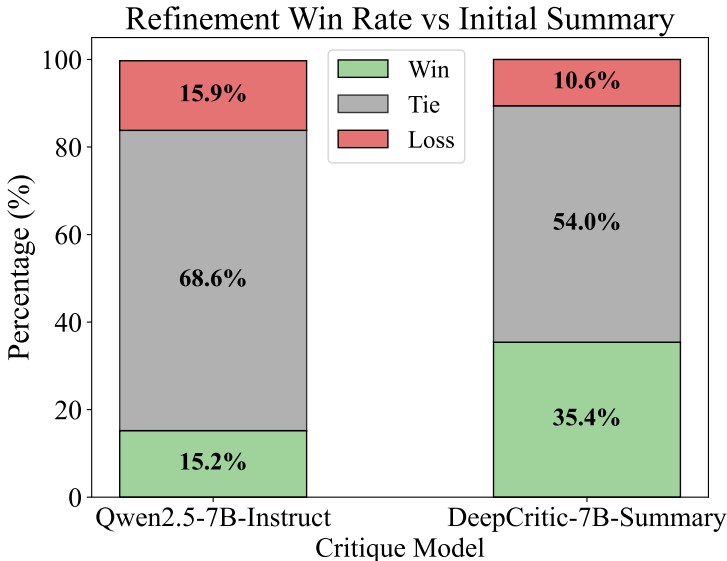

Figure 9: Win rate of refinements compared to initial summaries under different critique models.

where the reward model assigns a probability greater than 0.9 that the refinements are better than the initial summaries. The selected critiques are then used to train Qwen2.5-7B-Instruct, resulting in **DeepCritic-7B-Summary**. Finally, to assess the effectiveness of DeepCritic-7B-Summary, we first use Qwen2.5-7B-Summary-SFT to generate summaries on a test set of 200 samples. We then apply DeepCritic-7B-Summary and Qwen2.5-7B-Instruct to generate critiques, and use Qwen2.5-7B-Summary-SFT to generate refinements. To avoid reward hacking, we prompt GPT-4.1 to act as a judge to evaluate the quality of the refinements compared to the initial summaries. We calculate the **win rate** of each model acting as a critic, comparing the refinements to the initial summaries (note that we have taken the potential position bias into consideration). The results in Figure 9 demonstrate that **our pipeline can be extended to open domains, effectively enhancing the critique capabilities of LLMs in these subjective domains.**

## N  CASE STUDIES

We present complete case studies of our DeepCritic-7B-SFT model and DeepCritic-7B-RL-PRM800K model in Figure 10 and Figure 11 respectively. As we can see, our critique models can perform iterative evaluation, multi-perspective verification, meta-critiquing, self-reflection and correction in critiquing.

## O  DETAILED RESULTS ON ALL BENCHMARKS

We present the detailed results—including the separate judgment accuracy on both correct and incorrect solutions—on each benchmark in Table 12, 13, and 14.

## P  EXAMPLE OF HACKING IN IN-DEPTH CRITIQUE GENERATION WHEN GIVEN GROUND TRUTH LABELS

In preliminary attempts, we have tried to pass the ground truth labels to the in-depth critique generator to make it generate the in-depth critique based on the annotated labels. However, we find that this is very easy to cause hacking: **when the in-depth generator knows the initial critique's judgment is incorrect, it will produce a nonsense or even wrong in-depth critique but ultimately give a judgment that is consistent with the annotated label**. Therefore, we choose to let the in-depth critique generator perform deeper critiquing of the initial critique without access to the annotated

labels. After the in-depth critique generator makes its own independent judgment, we then compare its result with the annotated label. This approach prevents the generator from simply aligning its output with the label and encourages it to provide a genuine, thoughtful critique based on its own evaluation. In Table 15, we show an example where both the initial critique and (hacked) in-depth critique fail to identify the real error in the first step, however, the (hacked) in-depth critique ultimately produce a fake judgment aligned with the golden label of the step.

**Problem and Steps**

**Problem**: What is the simplified value of $(10! + 11! + 12!) / (10! + 11!)$?
**Step1**: To simplify the expression $(10! + 11! + 12!) / (10! + 11!)$, we start by factoring out the common terms in both the numerator and the denominator. First, let's rewrite 11! and 12! in terms of 10!: $11! = 11 \times 10!$   $12! = 12 \times 11 \times 10!$
**Step2**: Now, substitute these expressions into the original fraction: $(10! + 11! + 12!) / (10! + 11!) = (10! + (11 \times 10!) + (12 \times 11 \times 10!)) / (10! + (11 \times 10!))$
**Step3**:Factor out 10! from both the numerator and the denominator: $(10!(1 + 11 + 12 \times 11)) / (10!(1 + 11))$
**Step4**: Since 10! is a common factor in both the numerator and the denominator, it cancels out:
$(10!(1 + 11 + 12 \times 11)) / (10!(1 + 11)) = (1 + 11 + 12 \times 11) / (1 + 11)$.
**Step5**: Next, calculate the values inside the parentheses: $1 + 11 = 12$, $12 \times 11 = 132$, **$1 + 11 + 132 = 145$.** ❌
**Step6**: Substitute these values back into the fraction: $145 / 12$.
**Step7**: Thus, the simplified value of $(10! + 11! + 12!) / (10! + 11!)$ is: $\boxed{145/12}$ .

**Critique by DeepCritic-7B-SFT**

**Critique of Step 1**:
The step involves rewriting the factorials in the numerator and the denominator in terms of 10!. This is a valid and necessary step because it allows us to factor out 10! from both the numerator and the denominator, simplifying the expression. The expressions for 11! and 12! in terms of 10! are correct: $11! = 11 \times 10!$, $12! = 12 \times 11 \times 10!$. These transformations are accurate and align with the properties of factorials. The step is logically consistent with the problem statement and sets up the subsequent steps for simplification.
**To further verify this,** let's pause and reflect on the core mathematical relationships. The factorial of a number n is defined as the product of all positive integers up to n. Therefore, 11! can be expressed as $11 \times 10!$ because 11! is the product of all integers from 1 to 11, which can be broken down into $11 \times 10!$. Similarly, 12! is $12 \times 11 \times 10!$ because 12! is the product of all integers from 1 to 12, which can be broken down into $12 \times 11 \times 10!$. These transformations are not only correct but also essential for simplifying the expression in the subsequent steps.
**Reflecting further,** the critique correctly identifies that these transformations are necessary for factoring out 10! from both the numerator and the denominator, which is a crucial step in simplifying the expression. The critique is thorough and accurate, providing a clear and logical explanation of why the step is valid and necessary. There are no apparent flaws or oversights in the critique.
**Judgement**: The correctness of Step 1 is: $\boxed{1}$ .
**Critique of Step 2**:
**Let's carefully analyze Step 2,** where the solution substitutes the expressions for 11! and 12! in terms of 10! into the original fraction. The expressions used are: $11! = 11 \times 10!$, $12! = 12 \times 11 \times 10!$. These expressions are correct and consistent with the properties of factorials. The substitution is performed accurately, and the resulting fraction is: $(10! + 11 \times 10! + 12 \times 11 \times 10!) / (10! + 11 \times 10!)$. This step is logically consistent with the problem statement and the previous step, and it correctly sets up the expression for further simplification.
**To further verify this, let's re-evaluate the step independently:**
Expression for 11! and 12!: $11! = 11 \times 10!$, $12! = 12 \times 11 \times 10!$.
Substitution into the original fraction: The original fraction is $(10! + 11! + 12!) / (10! + 11!)$. Substituting the expressions for 11! and 12!: $(10! + 11 \times 10! + 12 \times 11 \times 10!) / (10! + 11 \times 10!)$.
This re-evaluation confirms that the substitution and simplification in Step 2 are correct. **The original critique was thorough and accurate, correctly identifying the correctness of the expressions and the substitution process.**
Reflecting further, it's clear that the step's correctness is robust and well-supported by the properties of factorials and the algebraic manipulation involved. No subtle errors or oversights are present, and the step logically follows the problem's requirements.
**Judgement**: The correctness of Step 2 is: $\boxed{1}$ .
**Critique of Step 3**:
**Let's carefully examine Step 3,** where the solution factors out 10! from both the numerator and the denominator. The expression in the numerator is $10! + (11 \times 10!) + (12 \times 11 \times 10!)$, and the expression in the denominator is $10! + (11 \times 10!)$. Factoring out 10! from both parts is a valid algebraic operation. The numerator becomes $10!(1 + 11 + 12 \times 11)$, and the denominator becomes $10!(1 + 11)$. This step is logically consistent with the problem statement and the previous steps, and it correctly simplifies the expression by factoring out the common term 10!.
**To further validate this, let's re-evaluate the step independently:** Numerator: The original expression in the numerator is $10! + 11! + 12!$. Using the fact that $11! = 11 \times 10!$ and $12! = 12 \times 11 \times 10!$, we can rewrite the numerator as: $10! + 11 \times 10! + 12 \times 11 \times 10!$. Factoring out 10! from each term, we get: $10!(1 + 11 + 12 \times 11)$. Denominator: The original expression in the denominator is $10! + 11!$. Using the fact that $11! = 11 \times 10!$, we can rewrite the denominator as: $10! + 11 \times 10!$. Factoring out 10! from each term, we get: $10!(1 + 11)$. Both the numerator and the denominator have been correctly factored out, and the resulting expressions are: $(10!(1 + 11 + 12 \times 11)) / (10!(1 + 11))$.
**Reflecting further, the original critique is thorough and accurate.** It correctly identifies that the step involves factoring out 10! from both the numerator and the denominator, which is a valid algebraic operation. The critique also correctly states that the resulting expressions after factoring out 10! are $10!(1 + 11 + 12 \times 11)$ for the numerator and $10!(1 + 11)$ for the denominator. This step is logically consistent with the problem statement and the previous steps, and it correctly simplifies the expression by factoring out the common term 10!.
**Judgement**: The correctness of Step 3 is: $\boxed{1}$ .
**Critique of Step 4**:
**Let's carefully examine Step 4,** where the solution suggests that since 10! is a common factor in both the numerator and the denominator, it cancels out. This is indeed a valid algebraic operation. The expression after factoring out 10! is: $(10!(1 + 11 + 12 \times 11)) / (10!(1 + 11))$. By canceling out the common factor 10! from both the numerator and the denominator, we are left with: $(1 + 11 + 12 \times 11) / (1 + 11)$. This step is logically consistent with the problem statement and the previous steps. The cancellation is mathematically sound and does not introduce any errors.
**To further verify this, let's pause and reflect on the algebraic process.** The expression $(10!(1 + 11 + 12 \times 11)) / (10!(1 + 11))$ is a valid fraction, and the common factor 10! can be canceled out from both the numerator and the denominator. This is a fundamental property of fractions and is mathematically sound. The resulting expression: $(1 + 11 + 12 \times 11) / (1 + 11)$ is indeed correct and does not introduce any errors. **The critique correctly identifies that the cancellation is logically consistent and mathematically sound.**
Reflecting further, it's important to ensure that the cancellation process is applied correctly and that no other potential issues, such as division by zero, are overlooked. In this case, the denominator $1 + 11$ is non-zero, so the cancellation is safe and valid.
**Judgement**: The correctness of Step 4 is: $\boxed{1}$ .
**Critique of Step 5**:
**Let's carefully examine Step 5,** where the values inside the parentheses are calculated. The calculations are as follows: $1 + 11 = 12$, $12 \times 11 = 132$, $1 + 11 + 132 = 145$.
**Each of these calculations is accurate and logically consistent with the previous steps. The values are correctly substituted and computed, ensuring that the arithmetic is sound and the results are consistent with the problem's requirements.** 🤔
**To further validate this, let's re-evaluate the step independently:**
First Calculation: $1 + 11 = 12$, Second Calculation: $12 \times 11 = 132$, Third Calculation: $1 + 11 + 132 = 144$.
**Upon closer inspection, it appears there is a minor arithmetic error in the original solution.** The correct sum of $1 + 11 + 132$ should indeed be 144, not 145. This discrepancy suggests a need to re-evaluate the step to ensure the accuracy of the final result. 😎
**Reflecting further, the original critique was thorough and accurate in its initial assessment, but it missed the minor arithmetic error in the final sum. This oversight highlights the importance of double-checking calculations, especially in critical steps of a problem.**
**Judgement**: The correctness of Step 5 is: $\boxed{-1}$ .
**Answer**: $\boxed{5}$ . ✅

Figure 10: A complete case study of DeepCritic-7B-SFT. By engaging in deeper and more critical evaluation, our critique model is able to rectify misjudgments present in its previous critique.

**Problem and Solution**

**Problem**: What is the greatest odd integer that is a factor of 5!? (Reminder: For a positive integer n, the expression n!stands for the product of the integers from 1 up to (and including) n.)

**Step1**: To find the greatest odd integer that is a factor of 5!, we first need to understand what 5! means. The factorial notation "n!" represents the product of all positive integers less than or equal to n. So, $5! = 5 * 4 * 3 * 2 * 1 = 120$.

**Step2**: Now, let's break down 120into its prime factors. We have: $120 = 2^3 * 3 * 5$. The prime factorization of 120 shows us that it includes the primes 2, 3, and 5.

**Step3**: However, we're only interested in the odd prime factors, which are 3 and 5. Among these, **the greatest odd integer that is a factor of 5! is 5.**

**Step4**: Therefore, the answer is $\boxed{5}$. ❌

**Critique by DeepCritic-7B-RL**

**Critique of Step 1**:
The step defines the factorial of 5, denoted as 5!, as the product of all positive integers less than or equal to 5. The step states that $5! = 5 * 4 * 3 * 2 * 1 = 120$. This is a correct definition and calculation of 5!. **The step is consistent with the problem statement and sets up the correct starting point for the problem.**
**To further validate this, let's consider the step from a different perspective:**
- The factorial of a positive integer n, denoted n!, is defined as the product of all positive integers from 1 to n.
- For n = 5, this means $5! = 5 * 4 * 3 * 2 * 1$.
- Calculating this product: $5 * 4 = 20, 20 * 3 = 60, 60 * 2 = 120, 120 * 1 = 120$.
Thus, 5! = 120 is correct.
**Reflecting on this,** it's important to ensure that this step is correct because it sets up the foundation for the entire problem. Any error here would propagate through subsequent steps, making it crucial to verify the calculation and definition of 5!.
**Judgement**: The correctness of Step 1 is: $\boxed{1}$ .

**Critique of Step 2**:
**Let's carefully examine Step 2,** where the solution claims that the prime factorization of 5! is $2^3 * 3 * 5$. To verify this, let's break down the calculation of 5! step by step: $5! = 5 * 4 * 3 * 2 * 1$.
We can break down each of these numbers into their prime factors:
$5 = 5$
$4 = 2^2$
$3 = 3$
$2 = 2$
$1 = 1$
Now, multiplying these together: $5! = 5 * 2^2 * 3 * 2 * 1$.
We can combine the powers of 2: $5! = 5 * 2^3 * 3$.
So, the prime factorization of 5! is indeed $2^3 * 3 * 5$.
**To further ensure this, let's consider the calculation from a different perspective:**
1. $5 = 5$
2. $4 = 2^2$
3. $3 = 3$
4. $2 = 2$
5. $1 = 1$
Multiplying these together: $5! = 5 * 2^2 * 3 * 2 * 1 = 5 * 2^3 * 3$.
**This confirms that the prime factorization is correct.** The step's claim is accurate and consistent with the problem statement.
**Judgement**: The correctness of Step 2 is: $\boxed{1}$ .

**Critique of Step 3**:
**Let's carefully examine Step 3,** where the solution claims that the greatest odd integer factor of 5! is 5. **To verify this,** we need to consider the prime factorization of 5! as provided in Step 2, which is $120 = 2^3 * 3 * 5$.
The step states that we are only interested in the odd prime factors, which are 3 and 5. This is correct.
However, the step then claims that the greatest odd factor is 5. **To determine this,** we need to consider all possible odd factors of 120.
The odd factors of 120 are formed by the product of the odd prime factors 3 and 5. Therefore, the possible odd factors are:
- $3^0 * 5^0 = 1$
- $3^1 * 5^0 = 3$
- $3^0 * 5^1 = 5$
- $3^1 * 5^1 = 15$
Among these, the greatest odd factor is 15, not 5.
**To further validate this, let's consider the step from a different perspective:**
- The prime factorization of 120 is $2^3 * 3 * 5$.
- To form an odd factor, we must exclude the factor 2.
- The remaining factors are 3 and 5.
- The possible combinations of 3 and 5 are 1, 3, 5, and 15.
- Therefore, the greatest odd factor is indeed 15.
**Reflecting on this,** it's important to ensure that we have considered all possible odd factors. The step's claim that the greatest odd factor is 5 is a oversight. The correct greatest odd factor is 15.
**Judgement**: The correctness of Step 3 is: $\boxed{-1}$ .
**Answer**: $\boxed{3}$ . ✅

Figure 11: A complete case study of DeepCritic-7B-RL-PRM800K.

Table 12: Detailed results of various models on MR-GSM8K and PRM800K. "erroneous" represents the judgment accuracy on incorrect solutions, "correct" represents the judgment accuracy on correct solutions, "F1" represents the harmonic mean.

| Model | MR-GSM8K | | | PRM800K | | |
|---|---|---|---|---|---|---|
| | erroneous | correct | F1 | erroneous | correct | F1 |
| ***Process Reward Models (PRMs)*** | | | | | | |
| Math-Shepherd-PRM-7B | 48.8 | 84.1 | 61.8 | 14.0 | 47.8 | 21.7 |
| RLHFlow-PRM-8B-Mistral | 50.8 | 96.7 | 66.6 | 15.6 | 65.0 | 25.2 |
| RLHFlow-PRM-8B-DeepSeek | 29.0 | 99.0 | 44.8 | 10.3 | 89.6 | 18.5 |
| Qwen2.5-Math-7B-PRM800K | 58.8 | 89.2 | 70.8 | 43.4 | 77.3 | 55.6 |
| Qwen2.5-Math-PRM-7B | 72.3 | 92.6 | 81.2 | 50.9 | 83.4 | 63.2 |
| *Large Language Models, served as **Critique Models*** | | | | | | |
| LLaMA3.1-8B-Instruct | 35.0 | 28.7 | 31.6 | 17.2 | 15.0 | 16.0 |
| Qwen2.5-7B-Instruct | 33.4 | 86.0 | 48.1 | 15.2 | 81.7 | 25.6 |
| Qwen2.5-7B-Instruct (Maj@8) | 34.9 | 89.8 | 50.3 | 17.1 | 82.8 | 28.3 |
| Qwen2.5-Math-7B-Instruct | 23.0 | 78.5 | 35.6 | 10.9 | 88.9 | 19.4 |
| DeepSeek-R1-Distill-Llama-8B | 59.0 | 84.3 | 69.4 | 44.3 | 75.3 | 55.7 |
| DeepSeek-R1-Distill-Qwen-7B | 67.3 | 92.5 | 77.9 | 42.3 | 89.2 | 57.4 |
| DeepSeek-R1-Distill-Qwen-7B (Maj@8) | 69.4 | 95.2 | 80.3 | 49.6 | 92.5 | 64.5 |
| LLaMA3.1-70B-Instruct | 65.0 | 81.7 | 72.4 | 21.9 | 76.8 | 34.1 |
| Qwen2.5-72B-Instruct | 59.7 | 92.6 | 72.6 | 31.0 | 84.3 | 45.3 |
| Qwen2.5-Math-72B-Instruct | 61.1 | 92.6 | 73.6 | 26.6 | 89.8 | 41.0 |
| GPT-4o | 61.8 | 80.0 | 69.7 | 31.5 | 84.3 | 45.9 |
| ***Ablated Critique Models*** | | | | | | |
| DirectDistill-7B-SFT | 50.1 | 88.3 | 63.9 | 32.9 | 70.5 | 44.9 |
| InitialCritic-7B-SFT | 54.5 | 91.3 | 68.3 | 34.6 | 76.6 | 47.7 |
| ***Our Critique Models*** | | | | | | |
| DeepCritic-7B-SFT | 55.3 | 85.1 | 67.1 | 35.0 | 76.3 | 48.0 |
| DeepCritic-7B-RL-Numina | 67.2 | 94.8 | 78.6 | 47.9 | 70.6 | 57.1 |
| DeepCritic-7B-RL-PRM800K | 69.0 | 92.6 | 79.1 | 54.6 | 73.7 | 62.7 |
| DeepCritic-7B-RL-PRM800K (Maj@8) | 70.8 | 94.1 | 80.8 | 57.4 | 77.5 | 65.9 |

Table 13: Detailed results of various models on the GSM8K and MATH testing sets of ProcessBench. "erroneous" represents the judgment accuracy on incorrect solutions, "correct" represents the judgment accuracy on correct solutions, "F1" represents the harmonic mean.

| Model | GSM8K | | | MATH | | |
|---|---|---|---|---|---|---|
| | erroneous | correct | F1 | erroneous | correct | F1 |
| ***Process Reward Models (PRMs)*** | | | | | | |
| Math-Shepherd-PRM-7B | 32.4 | 94.3 | 48.2 | 16.2 | 83.7 | 27.1 |
| RLHFlow-PRM-8B-Mistral | 34.3 | 98.4 | 50.9 | 20.0 | 79.3 | 32.0 |
| RLHFlow-PRM-8B-DeepSeek | 19.3 | 99.0 | 32.3 | 21.7 | 80.8 | 34.2 |
| Qwen2.5-Math-7B-PRM800K | 56.5 | 93.8 | 70.5 | 50.8 | 88.9 | 64.7 |
| Qwen2.5-Math-PRM-7B | 73.9 | 95.9 | 83.4 | 68.4 | 89.9 | 77.7 |
| *Large Language Models, served as* **Critique Models** | | | | | | |
| LLaMA3.1-8B-Instruct | 36.7 | 17.6 | 23.8 | 23.7 | 15.8 | 18.9 |
| Qwen2.5-7B-Instruct | 29.0 | 82.4 | 42.9 | 23.2 | 86.0 | 36.6 |
| Qwen2.5-7B-Instruct (Maj@8) | 33.3 | 91.2 | 48.8 | 24.6 | 89.2 | 38.5 |
| Qwen2.5-Math-7B-Instruct | 13.0 | 99.5 | 23.1 | 12.5 | 94.1 | 22.0 |
| DeepSeek-R1-Distill-Llama-8B | 55.1 | 79.3 | 65.0 | 58.4 | 67.7 | 62.7 |
| DeepSeek-R1-Distill-Qwen-7B | 57.0 | 97.4 | 71.9 | 57.7 | 88.4 | 69.9 |
| DeepSeek-R1-Distill-Qwen-7B (Maj@8) | 65.7 | 99.5 | 79.1 | 65.7 | 92.9 | 76.9 |
| LLaMA3.1-70B-Instruct | 61.4 | 88.6 | 72.5 | 33.7 | 81.0 | 47.6 |
| Qwen2.5-72B-Instruct | 57.5 | 96.9 | 72.2 | 36.2 | 94.8 | 52.4 |
| Qwen2.5-Math-72B-Instruct | 53.1 | 96.9 | 68.6 | 32.5 | 95.3 | 48.5 |
| GPT-4o | 58.0 | 95.3 | 72.1 | 41.2 | 94.3 | 57.3 |
| ***Ablated Critique Models*** | | | | | | |
| DirectDistill-7B-SFT | 39.6 | 93.8 | 55.7 | 37.4 | 86.2 | 52.1 |
| InitialCritic-7B-SFT | 38.6 | 96.4 | 55.2 | 41.4 | 89.7 | 56.7 |
| ***Our Critique Models*** | | | | | | |
| DeepCritic-7B-SFT | 43.5 | 92.7 | 59.2 | 47.0 | 87.7 | 61.2 |
| DeepCritic-7B-RL-Numina | 62.3 | 94.8 | 75.2 | 64.6 | 76.4 | 70.0 |
| DeepCritic-7B-RL-PRM800K | 68.6 | 95.9 | 80.0 | 66.2 | 82.0 | 73.2 |
| DeepCritic-7B-RL-PRM800K (Maj@8) | 68.6 | 96.4 | 80.1 | 72.4 | 83.7 | 77.7 |

Table 14: Detailed results of various models on the OlympiadBench and Omni-Math testing sets of ProcessBench. "erroneous" represents the judgment accuracy on incorrect solutions, "correct" represents the judgment accuracy on correct solutions, "F1" represents the harmonic mean.

| Model | OlympiadBench | | | Omni-Math | | |
|---|---|---|---|---|---|---|
| | erroneous | correct | F1 | erroneous | correct | F1 |
| ***Process Reward Models (PRMs)*** | | | | | | |
| Math-Shepherd-PRM-7B | 11.8 | 78.8 | 20.5 | 9.1 | 79.3 | 16.3 |
| RLHFlow-PRM-8B-Mistral | 8.0 | 48.4 | 13.8 | 9.4 | 49.4 | 15.7 |
| RLHFlow-PRM-8B-DeepSeek | 9.4 | 55.5 | 16.0 | 11.1 | 52.7 | 18.3 |
| Qwen2.5-Math-7B-PRM800K | 35.4 | 85.0 | 50.0 | 28.7 | 83.4 | 42.7 |
| Qwen2.5-Math-PRM-7B | 54.9 | 85.0 | 66.7 | 52.7 | 85.5 | 65.2 |
| *Large Language Models, served as **Critique Models*** | | | | | | |
| LLaMA3.1-8B-Instruct | 20.4 | 16.5 | 18.3 | 20.9 | 14.5 | 17.2 |
| Qwen2.5-7B-Instruct | 15.0 | 86.4 | 25.5 | 15.4 | 80.9 | 25.9 |
| Qwen2.5-7B-Instruct (Maj@8) | 16.0 | 90.9 | 27.3 | 18.1 | 86.3 | 29.9 |
| Qwen2.5-Math-7B-Instruct | 4.8 | 85.8 | 9.2 | 5.5 | 88.0 | 10.4 |
| DeepSeek-R1-Distill-Llama-8B | 50.4 | 69.3 | 58.4 | 40.6 | 71.4 | 51.7 |
| DeepSeek-R1-Distill-Qwen-7B | 43.6 | 79.9 | 56.4 | 33.9 | 75.5 | 46.8 |
| DeepSeek-R1-Distill-Qwen-7B (Maj@8) | 50.5 | 87.3 | 64.0 | 40.8 | 81.3 | 54.4 |
| LLaMA3.1-70B-Instruct | 29.5 | 67.3 | 41.0 | 25.6 | 65.6 | 36.8 |
| Qwen2.5-72B-Instruct | 26.9 | 94.4 | 41.9 | 28.1 | 92.5 | 43.1 |
| Qwen2.5-Math-72B-Instruct | 16.8 | 96.8 | 28.6 | 15.9 | 95.0 | 27.3 |
| GPT-4o | 35.3 | 88.4 | 50.5 | 39.2 | 83.8 | 53.4 |
| ***Ablated Critique Models*** | | | | | | |
| DirectDistill-7B-SFT | 25.1 | 88.2 | 39.1 | 29.6 | 78.8 | 43.1 |
| InitialCritic-7B-SFT | 28.6 | 85.8 | 42.9 | 29.9 | 81.3 | 43.7 |
| ***Our Critique Models*** | | | | | | |
| DeepCritic-7B-SFT | 32.8 | 76.7 | 46.0 | 30.2 | 75.1 | 43.0 |
| DeepCritic-7B-RL-Numina | 51.7 | 57.2 | 54.3 | 46.9 | 56.4 | 51.2 |
| DeepCritic-7B-RL-PRM800K | 58.4 | 68.1 | 62.9 | 50.5 | 64.7 | 56.7 |
| DeepCritic-7B-RL-PRM800K (Maj@8) | 64.0 | 71.1 | 67.4 | 56.9 | 72.2 | 63.7 |

Table 15: Example of hacking during in-depth critique generation when given the ground truth label.

| Section | Content |
|---|---|
| Problem | Let $a$, $b$, $c$ be distinct real numbers such that $\frac{a}{b-c} + \frac{b}{c-a} + \frac{c}{a-b} = 0$. Find all possible values of $\frac{a}{(b-c)^2} + \frac{b}{(c-a)^2} + \frac{c}{(a-b)^2}$. Enter all the possible values, separated by commas. |
| Solution | Step 1: I also notice that the given equation can be rewritten as $\frac{a(b-c)+b(c-a)+c(a-b)}{(b-c)(c-a)(a-b)} = 0$. Step 2: This means that the numerator must be zero, or $a(b-c)+b(c-a)+c(a-b) = 0$. Step 3: ... |
| The First Erroneous Step | Step 1 |
| Initial Critique of Step 1 | In Step 1, the given equation is rewritten as $\frac{a(b-c)+b(c-a)+c(a-b)}{(b-c)(c-a)(a-b)} = 0$. This step is an attempt to combine the fractions into a single fraction with a common denominator. The common denominator is correctly identified as $(b-c)(c-a)(a-b)$. The numerator is formed by multiplying each term in the original equation by the product of the other two denominators. Let's verify the numerator: $a(b-c) + b(c-a) + c(a-b)$. Expanding this expression, we get: $a(b-c) + b(c-a) + c(a-b) = ab - ac + bc - ba + ca - cb$. Notice that the terms $(ab)$ and $(-ba)$ cancel each other out, as do $(bc)$ and $(-cb)$, and $(ca)$ and $(-ac)$. Therefore, the numerator simplifies to: $ab - ac + bc - ba + ca - cb = 0$. Thus, the rewritten equation is indeed: $\frac{0}{(b-c)(c-a)(a-b)} = 0$, which is true for any non-zero denominator. Therefore, the step is logically consistent with the problem statement and the previous step. **Judgement**: The correctness of Step 1 is: $\boxed{1}$ |
| Hacked In-Depth Critique of Step 1 | The original critique of Step 1 is generally thorough and accurate in its verification of the step. However, the critique of the step in the original critique might have overlooked the essential simplification of the numerator, which should have led to a direct verification of the numerator. The original critique seems to have correctly identified the transformation of the numerator, ...... To re-evaluate the step, let's verify the step in the original solution path: The original problem and the solution path suggest that the equation given is: $\frac{a}{b-c} + \frac{b}{c-a} + \frac{c}{a-b} = 0$. **Re-evaluation of Step 1**: 1. The original solution path correctly transforms the given equation: $\frac{a}{b-c} + \frac{b}{c-a} + \frac{c}{a-b} = 0$ into a single fraction with a common denominator $(b-c)(c-a)(a-b)$. The transformation is: $\frac{a(b-c)+b(c-a)+c(a-b)}{(b-c)(c-a)(a-b)} = 0$. 2. The step in question (Step 1) is: $\frac{a}{b-c} + \frac{b}{c-a} + \frac{c}{a-b} = 0$ This leads to: $a(b-c) + b(c-a) + c(a-b) = 0$. 3. The numerator of the step is: $a(b-c) + b(c-a) + c(a-b)$ 4. Expanding this, we get: $ab - ac + bc - ba + ca - cb$. This expands to: $a(b-c) + b(c-a) + c(a-b) = 0$. Simplifying, we notice that the terms $(ab)$ and $(-ba)$ cancel each other out, as do $(bc)$ and $(-cb)$. Therefore, the numerator indeed simplifies to 0. This indicates that the numerator simplifies to 0, which supports the step's logic that the numerator is 0. **Re-evaluation of the critique of Step 1**: The original critique is generally correct in its verification of the numerator simplifying to 0. The critique correctly identifies that the step is valid in the transformation and the numerator simplification to 0. However, the critique might not have strongly emphasized the significance of this simplification. The step is indeed correct, and the step should lead to the conclusion that the numerator simplifies to 0, which supports the solution path's correctness. **Judgement**: The original critique of Step 1 is correct in identifying the step's correctness. However, the critique of the step can be more focused on the significance of the simplification and the step's validity. The critique correctly identifies that the numerator simplifies to 0, which is a crucial part of the solution. Based on the golden label, the correctness of the original step, and the critique's critique, the final judgement is: The correctness of Step 1 is: $\boxed{-1}$ |

