# OpenReview forum: "DeepCritic: Deliberate Critique with Large Language Models"
_ICLR.cc/2026/Conference — Submitted to ICLR 2026_

### Official Review · Reviewer_y2i7 · 2025-10-24

**Soundness:** 3
**Presentation:** 3
**Contribution:** 3
**Rating:** 6
**Confidence:** 4

**Summary:**

This paper addresses a critical and timely issue: the superficial nature of critiques generated by Large Language Models (LLMs) when serving as evaluators, particularly in complex reasoning domains such as mathematics. The authors argue that existing LLM critics often produce shallow feedback that merely echoes the original solution's reasoning, leading to low error detection accuracy and insufficient guidance for correction.

To overcome this limitation, the paper introduces DeepCritic, a novel two-stage training framework designed to teach LLMs to perform "deliberate critique."

Stage 1 (Supervised Fine-Tuning): The core innovation lies in its sophisticated data curation pipeline. First, a powerful teacher model (Qwen2.5-72B-Instruct) generates an "initial critique" for each step of a mathematical solution. Crucially, the teacher model then performs a deeper "meta-critique" on the initial critique itself. This involves multi-perspective verification and self-reflection to identify shortcomings in the initial assessment. Finally, these two critiques are synthesized into a single, high-quality, long-form "deliberate critique." A curated dataset of 4.5K such examples is used to fine-tune the base model.

Stage 2 (Reinforcement Learning): Building upon the SFT model, RL is employed to further enhance its critique capabilities. The reward signals are sourced from two pathways: 1) existing human-annotated datasets (e.g., PRM800K), and 2) automatically generated step-level labels via Monte Carlo sampling, enabling scalable oversight.

Experimental results demonstrate that the resulting DeepCritic-7B model significantly outperforms much larger and more capable baselines, including GPT-4o, on several mathematical error detection benchmarks. The paper further showcases the model's practical utility as both a verifier to improve solution accuracy and as a supervisor to guide other LLMs in refining their errors, even demonstrating a potential for weak-to-strong supervision.

**Strengths:**

- Importance and Timeliness of the Problem: The paper tackles a fundamental challenge at the forefront of LLM development. As the community shifts from outcome-based to process-based supervision, improving the quality of automated feedback (i.e., critique) is paramount for achieving scalable oversight and building more reliable and trustworthy LLMs. This work directly addresses a core bottleneck in this research direction.

- Novel and Insightful Methodology: The paper's primary contribution is its ingenious "meta-critique" data generation strategy. This "critique of the critique" design cleverly pushes the model beyond simple right/wrong judgments, training it to perform multi-perspective verification, reflection, and self-correction. This is a far more profound approach than standard knowledge distillation, as it aims to instill a pattern of critical reasoning rather than just transferring knowledge.

- Strong and Comprehensive Experimental Validation: The empirical results are compelling and well-executed.
	- The performance is impressive, with a 7B model outperforming top-tier models like GPT-4o on specialized tasks, clearly demonstrating the method's efficacy.
	- The evaluation is thorough, conducted across multiple standard benchmarks and supported by detailed ablation studies that justify the necessity of each component of the framework.
	- The paper effectively demonstrates the model's practical potential beyond benchmark scores, showcasing its value in applied scenarios like verified majority voting and solution refinement, with the latter hinting at the exciting prospect of weak-to-strong supervision.

**Weaknesses:**

- Scalability and Cost of the Data Generation Pipeline: The framework's main strength—its high-quality data—is also a potential weakness in terms of scalability. The data curation process is computationally intensive, requiring multiple long-sequence inference passes from a very large teacher model for each data point. This makes the cost of data generation exceedingly high, posing a significant challenge for scaling the dataset to millions of examples and potentially limiting its feasibility for large-scale industrial deployment.

- Generalization to Other Domains: The method's success is demonstrated convincingly in the mathematical domain, which benefits from objective and verifiable ground truths. However, its generalizability to more subjective domains remains an open question.

- High Dependency on the Teacher Model: The capabilities of the resulting DeepCritic model are inherently capped by the proficiency of the teacher model. Any systematic biases, knowledge gaps, or reasoning flaws present in the teacher are likely to be inherited, and possibly amplified, through the data generation process. The paper does not explore how the choice and capability of the teacher model impact the final outcome.

- Ambiguity in Direct Application as an RL Reward Model: While RL is used in the second stage of training, the DeepCritic model itself does not produce a scalar reward suitable for direct use in standard online RL algorithms (e.g., PPO). Its output is a long-form text, making it challenging to efficiently integrate as a reward function in a closed-loop training setup. Its more immediate application appears to be as an offline data generator for preference tuning (e.g., DPO) or as a test-time refinement mechanism, while its role in online RL remains less clear.

**Questions:**

See Weakness

---

> ### Author Response · Authors · 2025-11-21
> **Author Response (Part 1)**
>
> We sincerely appreciate your positive and constructive review of our manuscript. We are encouraged that you think the problem we address to be both critical and timely, and we are particularly grateful for your recognition of our methodological innovation, especially the "meta-critiquing" data curation strategy. We appreciate your positive assessment of the effectiveness and comprehensiveness of our experimental validation, acknowledging that the results are "compelling and well-executed," is highly motivating. We make the following response to address your remaining questions.
>
> -----
>
>
> **Q1:** Regarding the scalability and cost of the data generation pipeline.
>
> **A1:** We would like to emphasize that **our data construction pipeline has good scalablility**. First, only a small amount of SFT data is required for critique teaching, since this stage primarily serves to demonstrate the desired deliberate critiquing format rather than to provide extensive supervision. Second, the proposed pipeline for automatically constructing RL data also shows good scalability. In our completion-rollout stage, we only use a 7B-sized generator (i.e., Qwen2.5-7B-Instruct). Even with the very strict filtering criteria described in Lines 274–281, the overall inference cost remains modest compared with the compute required by the subsequent RL training. The following Table 1 reports the estimated GPU hours for constructing the 4.5K SFT data, automatically curating the 14K RL data, and performing RL on this 14K dataset, respectively. As shown, **the computational overhead of data construction is smaller than the RL training cost, indicating that our data construction pipeline is indeed scalable**.
>
> More importantly, our method avoids the reliance on human-annotated datasets such as PRM800K. **The cost of using a model to automatically annotate data is significantly lower than manual labeling, making our approach a promising direction for achieving scalable oversight.**
>
> Furthermore, **our rigorous filtering procedures ensure very high data quality for RL**, which in turn leads to more effective and stable RL training.
>
> Table 1. Consumed GPU Hours in each stage of our method.
> |4.5K SFT Data Curation with 72B Teacher | 14K RL Data Curation with 7B Generator | RL on 14K Data with 7B Model|
> |:----:|:---:|:----:|
> | ~350 |~150  |  ~550 |
>
> -----
>
>
> **Q2:** Regarding the generalization of our approach to domains beyond mathematics.
>
> **A2:** Thank you for your question. First, in our submission, we have validated the effectiveness of our DeepCritic pipeline on verifiable domains (i.e., mathematics), which play a crucial role in enabling the effective advancement of the current RLVR (Reinforcement Learning with Verifiable Rewards) research landscape. Moreover, the mathematical reasoning domain has also been extensively examined in prior critique-enhancement studies [1,2,3]. Therefore, **our investigation into improving the critique capabilities of LLMs in verifiable domains carries substantial significance for advancing scalable oversight**.
>
> Then, we would like to clarify that **our framework can be naturally extended to other domains**. For subjective tasks, one can leverage LLM generators to evaluate critique quality by comparing whether the refinement produced based on a critique model leads to a higher-quality response than the original one (using an LLM-as-a-Judge mechanism). This quality comparison can then be used to assign different rewards to critiques and train the critique model via RL.
>
> We are following your suggestion to conduct additional experiments on more domains. We will update the new results once the experiments are done in the next few days.
>
> -----
>
> [1] Zheng, Xin, et al. "Critic-cot: Boosting the reasoning abilities of large language model via chain-of-thought critic." ACL 2025 Findings
>
> [2] Gao, Bofei, et al. "Llm critics help catch bugs in mathematics: Towards a better mathematical verifier with natural language feedback." ACL 2025 Findings
>
> [3] Tang, Zhengyang, et al. "Self-Evolving Critique Abilities in Large Language Models." COLM 2025

---

> ### Author Response · Authors · 2025-11-21
> **Part 2**
>
> **Q3:** Regarding the dependency on the teacher model's capabilities.
>
> **A3:** We would like to point out that our method can be applied for **self-improvement of LLMs without requiring stronger teacher models**. To validate this, we conduct experiments by leveraging Qwen2.5-7B-Instruct itself for curating its own deliberate seed critique data by following the pipeline introduced in the paper. Then, we perform SFT and RL (with auto-constructed Numina-14K) on Qwen2.5-7B-Instruct under the same settings as in the main experiments. The self-improvement results in the following table **demonstrate the effectiveness and promise of our method for LLM self-improvement**.
>
>
> Table 2. The results of self-improvement experiments on Qwen2.5-7B-Instruct. Values in parentheses indicate the improvement over the base model.
>
> |  | MR-GSM8K| PRM800K| PB-GSM8K| PB-MATH | PB-OlympiadBench | PB-OmniMath | Avg. |
> |--------|:----------:|:------:|:------:|:------:|:----:|:------:|:------:|
> | Qwen2.5-7B-Instruct  |  48.1 | 25.6 | 42.9 | 36.6 | 25.5 | 25.9 | 34.1 |
> | DeepCritic-7B-SFT (Self-Improvement)  |  51.2 (+3.1)  | 33.3 (+7.7) | 44.4 (+1.5) | 41.1 (+4.5) | 28.9 (+3.4) | 30.1 (+4.2) | 38.2 (+4.1)  |
> | DeepCritic-7B-RL-Numina (Self-Improvement)  |  **76.1** (+28.0)  | **49.6** (+24.0) | **68.8** (+25.9) | **62.9** (+26.3) | **51.2** (+25.7) | **48.8** (+22.9) | **59.6** (+25.5) |
>
> -----
>
> **Q4:** Regarding the discussion of DeepCritic's application as an RL reward model.
>
> **A4:** Thank you for this constructive question. In our submission, we have demonstrated the strong potential of DeepCritic for supervising model outputs at test time. Here, following your question, we further discuss two potential applications of DeepCritic during online RL training.
>
> (1) Standard RLVR (e.g., GRPO) penalizes an entire erroneous solution by reducing the advantages for all of its steps, including those that are actually correct, which may lead to sub-optimal learning efficiency. Since DeepCritic is able to identify the first erroneous step, incorporating DeepCritic into the RL pipeline would allow us to reassign advantages to the preceding correct steps more appropriately, thereby enabling the model to learn more effectively.
>
> (2) Second, DeepCritic can provide effective online feedback to help the policy model correct errors in its generated wrong solutions, enabling it to produce correct refinements especially for problems that it had never solved before during RL, thereby substantially improving training effectiveness. Existing works [4,5] have also demonstrated the benefits of such critique-driven feedback, and our critique models can provide a strong initialization for the critic components in their frameworks.
>
> We believe exploring the potential of incorporating DeepCritic into online RL training in above two aspects can be interesting and practical future directions.
>
> -----
>
> **We hope the above response addresses your questions. We are glad to continue the discussion if you have any further questions. We are incorporating above new results into the revision, and will update the submission once the revision is ready.**
>
>
> [4] Zhang, Xiaoying, et al. "Critique-grpo: Advancing llm reasoning with natural language and numerical feedback." arxiv 2025
>
> [5] Xi, Zhiheng, et al. "Critique-RL: Training Language Models for Critiquing through Two-Stage Reinforcement Learning." arxiv 2025

---

> > ### Author Response · Authors · 2025-11-27
> > **Looking forward to your feedback**
> >
> > Dear Reviewer y2i7,
> >
> > We sincerely thank you for reviewing our paper and providing helpful comments. We have addressed your questions in our rebuttal. We would like to know whether if you have any further questions, and we are glad to continue the discussion if there are any.
> >
> > We are sincerely looking forward to your feedback, and your support means a lot to us! Thank you.
> >
> > Best regards,
> >
> > Authors

---

> > > ### Author Response · Authors · 2025-12-02
> > > **New results**
> > >
> > > Dear Reviewer y2i7,
> > >
> > > In this message, we provide the additional results of applying our DeepCritic pipeline to a subjective domain, *text summarization*, to address you previous question about the generalizability of our method to open domains.
> > >
> > > We follow the previous study [1] to first use ```TL;DR``` dataset [2] to fine-tune Qwen2.5-7B-Instruct and get **Qwen2.5-7B-Summary-SFT**. We also train a BT reward model **Qwen2.5-7B-Summary-RM** on the provided preference data from [1]. Then, we leverage Qwen2.5-7B-Summary-SFT to generate summaries on a held-out set and follow the same iterative critique-generation pipeline outlined in the main paper to produce deliberate critiques for each summary. To filter useful critiques for training, we first use Qwen2.5-7B-Summary-SFT to refine summaries based on the corresponding critiques, and then score the initial summaries and their refinements using Qwen2.5-7B-Summary-RM. We filter out the critiques where the reward model assigns a probability greater than 0.9 that the refinements are better than the initial summaries. The selected critiques are then used to train Qwen2.5-7B-Instruct, resulting in **DeepCritic-7B-Summary**. Finally, to assess the effectiveness of DeepCritic-7B-Summary, we first use Qwen2.5-7B-Summary-SFT to generate summaries on a test set of 200 samples. We then apply DeepCritic-7B-Summary and Qwen2.5-7B-Instruct to generate critiques, and use Qwen2.5-7B-Summary-SFT to generate refinements. To avoid reward hacking, we prompt GPT-4.1 to act as a judge to evaluate the quality of the refinements compared to the initial summaries. We calculate the win rate of each model acting as a critic, comparing the refinements to the initial summaries (note that we have taken potential position bias into consideration). The results in the following table demonstrate that **our pipeline can be extended to open domains, effectively enhancing the critique capabilities of LLMs in these subjective domains.**
> > >
> > > Table 1. Win rate of refinements compared to initial summaries under different critique models.
> > >
> > > |  | Win | Tie | Loss |
> > > |--------|:--------:|:------:|:------:|
> > > |Qwen2.5-7B-Instruct  | 15.2\% | 68.6\% |  15.9\%|
> > > | DeepCritic-7B-Summary  |  **35.4\%** | **54.0\%** |  **10.6\%** |
> > >
> > >
> > >
> > > ----
> > >
> > > **The above new results along with the previous rebuttal have addressed all your remaining questions**, and we will incorporate these new results into the revision.
> > >
> > >
> > > [1] Stiennon, Nisan, et al. "Learning to summarize with human feedback."  NIPS 2020
> > >
> > > [2] Völske, Michael, et al. "Tl; dr: Mining reddit to learn automatic summarization." ACL 2017

---

### Official Review · Reviewer_PU76 · 2025-10-31

**Soundness:** 2
**Presentation:** 3
**Contribution:** 2
**Rating:** 4
**Confidence:** 4

**Summary:**

This paper introduces DeepCritic, a framework to enhance the critique capabilities of LLMs, specifically for mathematical reasoning. The authors address the superficial nature of current LLM critiques by proposing following the same stages (SFT + RL) as other reasoning task.
The resulting 7B-parameter critic model outperforms larger models (like GPT-4o) and specialized math models on error identification benchmarks. It also scales well at test-time and effectively guides generators to refine their answers.

**Strengths:**

1. Throughout experiments: the paper contains very throughout experiments to prove the idea. Although this is a well explored scope (SFT + RL), the full set of experiment is still nice.

2. The paper demonstrates that RL with automatically constructed data (DeepCritic-7B-RL-Numina) also yields substantial gains, this confirms with finding from other papers, and proving that auto rating is valuable.

**Weaknesses:**

1. Lack of novelty: the framework adopted in this paper is well established in reasoning world, this work can be viewed as an application in the critique capability.

2. Limited Domain: The paper focuses solely on mathematical reasoning. While a standard testbed, it's unclear if this deliberate critique approach generalizes well to more subjective or less structured domains (e.g., creative writing, complex instruction following).

3. Dependency on Strong Teacher: The seed data generation relies heavily on a very capable model (Qwen2.5-72B-Instruct). The approach might be less viable if a significantly stronger teacher model isn't available for a given domain. As we know that distill is very effective for reasoning, the result is kind depending on this.

**Questions:**

1. When you use the LLM as critic, what prompt was used? Have you tried to improve the prompt to improve the performance?
2. There are a lot mistakes in PRM800K (80% accuracy based on OpenAI), how does that affect the final result?

---

> ### Author Response · Authors · 2025-11-21
> **Author Response (Part 1)**
>
> We sincerely appreciate your thoughtful review and constructive feedback. We are encouraged that you find our experimental setup to be thorough and the results on automated data construction valuable. We address the following concerns and questions raised in your review.
>
> -----
>
> **Q1:** Regarding the novelty of the framework.
>
> **A1:** We would like to point out that the two-stage SFT-then-RL pipeline is a general training paradigm, but **we have introduced distinct and novel contributions in both stages compared with prior works**.
>
> First, in the SFT data curation, we introduce a novel in-depth critique generation stage. Our iterative data synthesis pipeline produces critique data **enriched with multi-perspective evaluation, meta-critiquing, and other deep analytical patterns**. In contrast to prior methods [1,2,3] that directly trains on initial critiques, our SFT data teaches the model to adopt more critical reasoning, employ more diverse verification strategies, and engage in self-reflection and self-correction during inference, ultimately enabling more advanced critiquing capabilities.
>
> Moreover, unlike prior studies [4,5,6] that rely solely on large-scale SFT to enhance critique capabilities, we take a pioneering step by introducing an RL phase that explicitly incentivizes step-wise critiquing abilities of LLMs. By reformulating the generative PRM task as predicting the index of the first erroneous step, we leverage the strengths and effectiveness of RLVR (Reinforcement Learning with Verifiable Rewards), **enabling the model to autonomously explore and unlock its latent potential, thereby achieving stronger generalization**.
>
> Thus, our work shows sufficient novelty and contribution.
>
> -----
>
>
> **Q2:** Regarding the generalization beyond mathematical reasoning to more subjective domains.
>
> **A2:** Thank you for your question. First, in our submission, we have validated the effectiveness of our DeepCritic pipeline on verifiable domains (i.e., mathematics), which play a crucial role in enabling the effective advancement of the current RLVR (Reinforcement Learning with Verifiable Rewards) research landscape. Moreover, the mathematical reasoning domain has also been extensively examined in prior critique-enhancement studies [4,5,6]. Therefore, **our investigation into improving the critique capabilities of LLMs in verifiable domains carries substantial significance for advancing scalable oversight**.
>
> Then, we would like to clarify that **our framework can be naturally extended to other domains**. For subjective tasks, one can leverage LLM generators to evaluate critique quality by comparing whether the refinement produced based on a critique model leads to a higher-quality response than the original one (using an LLM-as-a-Judge mechanism). This quality comparison can then be used to assign different rewards to critiques and train the critique model via RL.
>
> We are following your suggestion to conduct additional experiments on more domains. We will update the new results once the experiments are done in the next few days.
>
> -----
>
>
> [1] Saunders, William, et al. "Self-critiquing models for assisting human evaluators." arxiv 2022
>
> [2] Wang, Yubo, Xiang Yue, and Wenhu Chen. "Critique fine-tuning: Learning to critique is more effective than learning to imitate." arxiv 2025
>
> [3] Wang, Yubo, et al. "Unleashing the Reasoning Potential of Pre-trained LLMs by Critique Fine-Tuning on One Problem." EMNLP 2025
>
> [4] Zheng, Xin, et al. "Critic-cot: Boosting the reasoning abilities of large language model via chain-of-thought critic." ACL 2025 Findings
>
> [5] Gao, Bofei, et al. "Llm critics help catch bugs in mathematics: Towards a better mathematical verifier with natural language feedback." ACL 2025 Findings
>
> [6] Tang, Zhengyang, et al. "Self-Evolving Critique Abilities in Large Language Models." COLM 2025

---

> > ### Comment · Reviewer_PU76 · 2025-11-24
> >
> > For looking for the first error step, have you consider something similar to binary search algorithm like in this paper: https://arxiv.org/abs/2406.06592?

---

> > > ### Author Response · Authors · 2025-11-24
> > > **Reply**
> > >
> > > We appreciate your further feedback. Regarding the binary search algorithm you referred to in [1], we note that **although binary search improves the efficiency of search and filtering, the quality and accuracy of the resulting annotations are substantially inferior to those produced by our method, causing certain performance degradation in RL.**
> > >
> > > Binary search works by repeatedly selecting the midpoint of the remaining partial solution, performing a rollout from that point, and then determining whether the correct answer appears in the rollouts. If a correct answer exists, the first erroneous step must lie in the latter half; otherwise, it lies in the former half. This procedure avoids performing rollouts at every step, reducing the complexity from $\mathcal{O}(kN)$ to $\mathcal{O}(k\log N)$, where $N$ is the number of steps and $k$ is the number of rollouts. However, **this approach is vulnerable to both misclassification and missed detections.** Consider the scenario in which one of the rollouts performed at the midpoint produces a correct final answer: although the true first erroneous step may actually belong to the earlier half, a fortunate completion of the remaining steps during rollout erroneously signals that the error must lie in the latter half. Prior work [2] has shown that such misleading cases can constitute over 20% of all cases.
> > >
> > > In our method, we perform rollouts at every step of the solution and adopt **a much stricter criterion for identifying the first erroneous step**: it is defined as the first step from which, along with all subsequent steps, all rollouts generated by the generator are incorrect, while for all preceding steps, more than half of the rollouts reach the correct answer (as illustrated in Lines 274–281). Therefore, **the automatically annotated data produced by our approach is of substantially higher quality, which in turn enables RL to better exploit the model’s potential.**
> > >
> > > Then, we provide a preliminary experimental comparison. We compare the model trained using our strict data-annotation strategy (as adopted in the paper) with a model trained using RL data constructed under a weaker annotation criterion: when identifying the first erroneous step, we relax the requirement on its preceding steps: it suffices that, for each preceding step, at least one rollout reaches the correct answer. This setting can simulate the failure mode of binary search, where a prefix is incorrectly judged because a correct rollout happens to appear once by chance. The results in the table below show that **our stricter data-annotating strategy is crucial**.
> > >
> > > Table 1. Comparison results between weak and strict data-annotating strategies on ProcessBench.
> > >
> > > |  | GSM8K| MATH | OlympiadBench | OmniMath | Avg. |
> > > |--------|:-----:|:------:|:------:|:------:|:----:|
> > > | DeepCritic-7B-RL-Numina (weak criterion)   | 64.5 | 62.0 | 46.8 | 45.7 | 54.8 |
> > > | DeepCritic-7B-RL-Numina   | **75.2** | **70.0** | **54.3**  | **51.2** | **62.7** |
> > >
> > >
> > >
> > > -----
> > >
> > > We hope the above clarification solves your remaining question.
> > >
> > > [1] Luo, Liangchen, et al. "Improve mathematical reasoning in language models by automated process supervision."  arxiv 2024
> > >
> > > [2] Zheng, Chujie, et al. "Processbench: Identifying process errors in mathematical reasoning." ACL 2025

---

> > > > ### Comment · Reviewer_PU76 · 2025-11-24
> > > >
> > > > Good point about the random correct answer, with 20% error rate is acceptable, as even in PRM800K, human rating is with about 20% errors. I think one possible way is to raise the bar of correct half from 1 to 2. Based on my experience, the computation cost diff is too high, the error rate may be tolerable.

---

> > > > > ### Author Response · Authors · 2025-11-25
> > > > >
> > > > > Thank you for your further reply.
> > > > >
> > > > > We appreciate your helpful suggestion on balancing data-construction efficiency and RL performance. We agree that our noisy-label results indicate that binary search, when used with a relatively stricter filtering criterion, can be applied during data annotation to improve efficiency, which can further strengthen the practicability of our framework.
> > > > >
> > > > > Also, we report the GPU hours used for automatically curating the 14K RL data (first generate solutions, then perform Monte-Carlo sampling-based correctness estimation for all steps of all solutions, finally obtain 14K samples after filtering), and performing RL on this 14K dataset in the following table. As shown, even with our very strict data-annotation strategy, the computational overhead of data construction remains smaller than the subsequent RL training cost, indicating the scalability of our data construction pipeline. This may be because, when performing rollouts for each prefix, vLLM’s KV-cache mechanism accelerates repeated computations for instances sharing the same prefix.
> > > > >
> > > > > Table 1. Consumed GPU Hours in data-construction and RL stages.
> > > > > |14K RL Data Curation | RL on 14K Data|
> > > > > |:---:|:----:|
> > > > > |~150  |  ~550 |

---

> > > > > > ### Author Response · Authors · 2025-11-27
> > > > > > **Follow up**
> > > > > >
> > > > > > Dear Reviewer PU76,
> > > > > >
> > > > > > We sincerely thank you again for actively engaging in the discussion. We have attached the results of consumed GPU hours for RL data construction to show the scalability of our pipeline under the very strict data-filtering criterion. We further appreciate your insightful suggestion regarding the use of binary search, which could indeed enhance the practicality of our framework. We would like to know whether if you have any further questions, and we are glad to continue the discussion if there are any.
> > > > > >
> > > > > > We are sincerely looking forward to your further feedback, and your support means a lot to us! Thank you.
> > > > > >
> > > > > > Best regards,
> > > > > >
> > > > > > Authors

---

> > > > > > ### Comment · Reviewer_PU76 · 2025-11-27
> > > > > >
> > > > > > To be honest, the cost of RL training is larger than data curation is not a valid reason of not using more efficient algorithm.

---

> ### Author Response · Authors · 2025-11-21
> **Part 2**
>
> **Q3:** Regarding the dependency on a strong teacher model (e.g., Qwen2.5-72B-Instruct) for seed data generation.
>
> **A3:** We would like to point out that our method can be applied for **self-improvement of LLMs without requiring stronger teacher models**. To validate this, we conduct experiments by leveraging Qwen2.5-7B-Instruct itself for curating its own seed critique data by following the pipeline introduced in the paper. Then, we perform SFT and RL (with Numina-14K) on Qwen2.5-7B-Instruct under the same settings as in the main experiments. The self-improvement results in the following table **demonstrate the effectiveness and promise of our method for LLM self-improvement**.
>
>
> Table 1. The results of self-improvement experiments on Qwen2.5-7B-Instruct. Values in parentheses indicate the improvement over the base model Qwen2.5-7B-Instruct.
>
> |  | MR-GSM8K| PRM800K| PB-GSM8K| PB-MATH | PB-OlympiadBench | PB-OmniMath | Avg. |
> |--------|:-------:|:------:|:------:|:------:|:----:|:------:|:------:|
> | Qwen2.5-7B-Instruct  |  48.1 | 25.6 | 42.9 | 36.6 | 25.5 | 25.9 | 34.1 |
> | DeepCritic-7B-SFT (Self-Improvement)  |  51.2   (+3.1)  | 33.3 (+7.7) | 44.4 (+1.5) | 41.1 (+4.5) | 28.9 (+3.4) | 30.1 (+4.2) | 38.2 (+4.1)  |
> | DeepCritic-7B-RL-Numina (Self-Improvement)  |  **76.1** (+28.0)  | **49.6** (+24.0) | **68.8** (+25.9) | **62.9** (+26.3) | **51.2** (+25.7) | **48.8** (+22.9) | **59.6** (+25.5) |
>
> -----
>
> **Q4:** Regarding the prompt used for the LLM critic.
>
> **A4:** As explained in Line 317 and Line 815-817, the evaluation prompt is mainly based on the official prompt used in ProcessBench [7] but is further improved. We observe that the original prompt often causes small-sized models (e.g., Qwen2.5-7B-Instruct) to output the label directly without producing any intermediate chain of thoughts (CoTs), which substantially degrades their performance. To address this issue, we improve the prompt by explicitly instructing the model to first generate an evaluation CoT followed by the final judgment. This modification yields a clear performance boost for the baseline critique models. These explanations are in Appendix C.4, but we will incorporate them into the main content into the revision.
>
> -----
>
> **Q5:** Regarding the potential impact of label errors in the PRM800K dataset.
>
> **A5:** Thank you for this constructive question. First, the instances in PRM800K are all human-labeled, so the overall annotation quality is generally high (we are not aware of any source claiming that PRM800K contains ~20% label errors, we would be very grateful if you could share us some related references so we could include them in the discussion). However, we acknowledge that a certain amount of label noise is inevitable due to occasional ambiguous or noisy cases.
>
> To examine the impact of label noise on RL performance, we conduct additional experiments using **Numina-14K**, our curated dataset curated under stringent filtering procedures and therefore exhibiting reliable label accuracy. Specifically, we randomly corrupt 20% of the samples with wrong final answers in Numina-14K by modifying their labels (i.e., the first erroneous step's step index) to random incorrect values, producing **Numina-14K-Noisy**. We then perform RL on DeepCritic-7B-SFT using Numina-14K-Noisy, and put the comparison results in the following table. As we can see, **although the presence of noise in the RL data degrades the final performance slightly, the model is still able to improve through RL even under noisy conditions, demonstrating a certain degree of robustness**. The plausible explanation is that, in GRPO, if the label is corrupted and given the relatively large label space in our task, the rollout within that group are highly likely to all receive a reward of 0. As a result, the entire group does not contribute to the update, thereby avoiding any negative impact on the model (in few cases, if incorrect rollouts happen to produce answers that match the corrupted label, then the training can indeed be affected).
>
> Table 2. The results on noisy RL data.
>
>
> |  | MR-GSM8K| PRM800K| PB-GSM8K| PB-MATH | PB-OlympiadBench | PB-OmniMath | Avg. |
> |--------|:--------:|:------:|:------:|:------:|:----:|:------:|------:|
> | DeepCritic-7B-SFT  |   67.1 |  48.0 | 59.2 | 61.2 | 46.0 | 43.0 | 54.1  |
> | DeepCritic-7B-RL-Numina  |  **78.6** | **57.1** | **75.2** | **70.0** | 54.3  | **51.2** | **64.4** |
> | DeepCritic-7B-RL-Numina (20% Label Noise in Wrong Solutions)  | 76.4 | 55.7   | 72.1 | 68.1 | **55.8** | 50.3| 63.1 |
>
> -----
>
> **We hope the above response addresses your questions. We are glad to continue the discussion if you have any further questions. We are incorporating above new results into the revision, and will update the submission once the revision is ready.**
>
> [7] Zheng, Chujie, et al. "Processbench: Identifying process errors in mathematical reasoning." ACL 2025

---

> ### Author Response · Authors · 2025-12-02
> **New results**
>
> Dear Reviewer PU76,
>
> We sincerely thank you once again for your positive engagement in the discussion and for your valuable follow-up suggestion on improving the efficiency of label auto-annotation. Here, we provide additional experimental results in response to **your suggestion of using binary search [1] for auto-annotation**, as well as **your previous question regarding the generalizability of our method to subjective domains**.
>
> ---
>
> First, we have  followed your suggestion to use binary search for label auto-annotation. Specifically, the criterion for determining whether the previous half contains an erroneous step is as follows: if more than half of the rollouts sampled from the middle step are correct  (consistent with the criterion used in our main paper for assessing the correctness of a step), then the previous half is considered correct, and the first erroneous step appears in the subsequent half. We then curate 14K data, with a size and label distribution comparable to those used in our main experiments, for RL. The evaluation results are in the following table. We can observe that **using binary search may slightly degrade performance due to the larger label noise introduced by a weaker filtering criterion, but it significantly improves annotation efficiency**, reducing the complexity from $\mathcal{O} (N)$ to $\mathcal{O} (\log N)$ (where $N$ is the number of reasoning steps). Your suggestion helps to enhance the scalability of our framework.
>
> Table 1. Results of using binary search in label auto-annotation.
>
>
> |  | MR-GSM8K| PRM800K| PB-GSM8K| PB-MATH | PB-OlympiadBench | PB-OmniMath | Avg. |
> |--------|:--------:|:------:|:------:|:------:|:----:|:------:|------:|
> | DeepCritic-7B-SFT  |   67.1 |  48.0 | 59.2 | 61.2 | 46.0 | 43.0 | 54.1  |
> | DeepCritic-7B-RL-Numina  |  **78.6** | **57.1** | **75.2** | **70.0** | 54.3  | 51.2 | **64.4** |
> | DeepCritic-7B-RL-Numina (binary search)  | 77.0 | 56.0   | 72.1 | 67.5 | **57.0** | **53.8** | 63.9 |
>
> ---
>
> Then, we provide the additional results of applying our DeepCritic pipeline to a subjective domain, *text summarization*, to address you previous question about the generalizability of our method to open domains. We follow the previous study [2] to first use ```TL;DR``` dataset [3] to fine-tune Qwen2.5-7B-Instruct and get **Qwen2.5-7B-Summary-SFT**. We also train a BT reward model **Qwen2.5-7B-Summary-RM** on the provided preference data from [2]. Then, we leverage Qwen2.5-7B-Summary-SFT to generate summaries on a held-out set and follow the same iterative critique-generation pipeline outlined in the main paper to produce deliberate critiques for each summary. To filter useful critiques for training, we first use Qwen2.5-7B-Summary-SFT to refine summaries based on the corresponding critiques, and then score the initial summaries and their refinements using Qwen2.5-7B-Summary-RM. We filter out the critiques where the reward model assigns a probability greater than 0.9 that the refinements are better than the initial summaries. The selected critiques are then used to train Qwen2.5-7B-Instruct, resulting in **DeepCritic-7B-Summary**. Finally, to assess the effectiveness of DeepCritic-7B-Summary, we first use Qwen2.5-7B-Summary-SFT to generate summaries on a test set of 200 samples. We then apply DeepCritic-7B-Summary and Qwen2.5-7B-Instruct to generate critiques, and use Qwen2.5-7B-Summary-SFT to generate refinements. To avoid reward hacking, we prompt GPT-4.1 to act as a judge to evaluate the quality of the refinements compared to the initial summaries. We calculate the win rate of each model acting as a critic, comparing the refinements to the initial summaries (note that we have taken potential position bias into consideration). The results in the following table demonstrate that **our pipeline can be extended to open domains, effectively enhancing the critique capabilities of LLMs in these subjective domains.**
>
> Table 2. Win rate of refinements compared to initial summaries under different critique models.
>
> |  | Win | Tie | Loss |
> |--------|:--------:|:------:|:------:|
> |Qwen2.5-7B-Instruct  | 15.2\% | 68.6\% |  15.9\%|
> | DeepCritic-7B-Summary  |  **35.4\%** | **54.0\%** |  **10.6\%** |
>
>
>
> ----
>
> **The above new results along with the previous rebuttal have addressed all your remaining questions**, and we will incorporate these new results into the revision.
>
> [1] Luo L, et al. Improve mathematical reasoning in language models by automated process supervision. arxiv 2024
>
> [2] Stiennon, Nisan, et al. "Learning to summarize with human feedback."  NIPS 2020
>
> [3] Völske, Michael, et al. "Tl; dr: Mining reddit to learn automatic summarization." ACL 2017

---

### Official Review · Reviewer_Err4 · 2025-10-31

**Soundness:** 2
**Presentation:** 3
**Contribution:** 2
**Rating:** 4
**Confidence:** 3

**Summary:**

This paper introduces DeepCritic, a two-stage framework for enhancing the critique capabilities of large language models (LLMs) in mathematical reasoning tasks. Existing LLM critics often produce shallow critiques, leading to low judgment accuracy and insufficient feedback for error correction. The authors curate 4.5K long-form critiques incorporating multi-perspective verification and meta-critiquing for supervised fine-tuning (SFT), followed by reinforcement learning (RL) using either human-annotated data (e.g., PRM800K) or automatically generated data via Monte Carlo sampling. Experiments on benchmarks like MR-GSM8K, PRM800K, and ProcessBench demonstrate that the resulting 7B-parameter model outperforms baselines including GPT-4o and DeepSeek-R1-Distill, while enabling better refinement of LLM generators through detailed feedback. Contributions include a novel data curation pipeline for deliberate critiquing and evidence of weak-to-strong supervision potential.

**Strengths:**

The paper innovatively addresses the superficiality of LLM critiques by introducing a two-stage pipeline that combines iterative critique generation (initial + in-depth meta-critiquing) with RL, creatively adapting Monte Carlo sampling for automated RL data in math domains, which extends prior work on scalable oversight (e.g., Saunders et al., 2022) to deliberate reasoning without relying solely on human labels.

The methodology is rigorously evaluated across multiple benchmarks, showing substantial improvements (e.g., 20-point F1 gains post-SFT), with ablation studies validating key components like in-depth critiquing; the use of diverse RL data sources (human vs. auto-generated) provides robust evidence of the framework's effectiveness and generalizability.

The writing is clear, with detailed prompts, data statistics, and case studies illustrating deliberate critiquing; significantly, it advances LLM self-evolution by demonstrating how enhanced critics can supervise stronger generators (e.g., 7B critic refining 72B outputs), offering practical insights for superalignment and automated feedback in complex reasoning tasks.

**Weaknesses:**

While RL improves performance, the auto-generated data via Monte Carlo sampling discards certain solutions (e.g., fully correct/incorrect ones), potentially introducing biases toward medium-difficulty problems; this could be quantified with diversity metrics to ensure the data represents a wide range of math complexities.

Test-time scaling results focus on majority voting and refinement, but lack comparisons with advanced baselines like outcome reward models (ORMs) or hybrid PRM-ORM setups, which might reveal limitations in handling very long reasoning chains or non-math domains.

**Questions:**

Given the emphasis on meta-critiquing in SFT data, could you elaborate on how often the model exhibits self-correction during inference (e.g., via quantitative analysis of generated critiques), and whether this transfers to out-of-distribution math problems like those in higher-level Olympiads? A response with additional metrics could strengthen claims of deliberate reasoning robustness.

The RL stage uses GRPO with a binary accuracy reward; how sensitive is performance to alternative reward designs, such as incorporating critique informativeness (e.g., via BLEU-like scores on feedback quality)? Experiments or ablation on this could address potential over-optimization toward judgment accuracy at the expense of feedback depth.

In weak-to-strong supervision experiments, the 7B critic refines 72B generators effectively, but what happens when the generator is even stronger (e.g., GPT-4o level) or in adversarial settings where solutions have subtle logical flaws? Providing case studies or extended results here could clarify the framework's limits in superalignment scenarios.

To better position your work in the literature, could you include a comparison with related critique enhancement methods such as CTRL, AlignRAG, Critique fine-tuning, one-shot CFT, and Critique-Guided Distillation? Specifically, discuss how your iterative meta-critiquing and RL pipeline differs in terms of data efficiency, critique depth, and applicability to math reasoning, potentially through qualitative or quantitative contrasts to highlight unique advantages or limitations?

---

> ### Author Response · Authors · 2025-11-21
> **Author Response (Part 1)**
>
> We sincerely appreciate your constructive feedback and thoughtful questions. We are encouraged that you find our work to propose an innovative framework for enhancing the critique capabilities of LLMs. We are glad that you think our approach has been rigorously validated through comprehensive experiments, which clearly demonstrate the effectiveness of our critique model in automated supervision. We address the following concerns and questions raised in your review.
>
>
> -----
>
> **Q1:** Regarding the concern on the potential difficulty bias in auto-generated RL data.
>
> **A1:** We would like to clarify that **our automatic data-construction pipeline does not introduce difficulty bias**.
>
> First, we collect and filter problems from the GSM8K, MATH, and Olympiads subsets of the NuminaMath dataset, ensuring **a broad coverage of problem difficulty levels**. Then, we use generators of different capabilities (Qwen2.5-1.5B/3B/7B-Instruct) to produce diverse solution attempts for each problem，which **increases the distributional variety of solutions**. In such cases, if all generated solutions for a given problem are correct, this indicates that the problem is relatively easy—even for the smallest 1.5B model. Consequently, the correctness of its solutions is trivial to determine (e.g., they may require only one or two reasoning steps). Conversely, if all solutions generated for a problem are incorrect, then even the 7B model consistently fails on it. In this scenario, when Qwen2.5-7B-Instruct is later used as the generator for Monte Carlo–based labeling, the resulting labels are not reliable (the auto-annotated label will always be the first step index), because the generator itself lacks the sufficient capability to solve the underlying problem. Therefore, **this filtering step is essential for ensuring the quality and trustworthiness of our RL dataset.**
>
> Moreover, we diaplay the statistics of problem sources in the resulting 14K instances in the following table to further **demonstrate the diverse distribution of problem difficulty in our final dataset**.
>
>
> Table 1. The statistics of problem sources in the auto-constructed 14K RL data.
>
> | Source | # Problems | Proportion (%) |
> |--------------|-----------:|---------------:|
> | GSM8K       |     5483  |      38.4%     |
> | MATH         |    5359   |      37.6%     |
> | Olympiads    |     3424   |      24.0%     |
> | **Total**    |   **14266**|    **100.0%**  |
>
>
> ------
> **Q2:** Regarding the comparison with advanced outcome/process reward models.
>
> **A2:** In our submission, we have demonstrated in Table 3 that **our critique model outperforms the state-of-the-art PRM baseline, ```Qwen2.5-Math-7B-PRM```, under the critique-based refinement setting**. Following your suggestion, we further compare ```Qwen2.5-Math-7B-PRM``` and a state-of-the-art 72B ORM, ```Qwen2.5-Math-RM-72B```, with our model under both majority-voting and critique-based refinement settings.
> The results in the following tables demonstrate **the great effectiveness of our method in enhancing the test-time scaling performance of the generator**. Specifically, our critique model can provide detailed and informative feedback on the generator’s outputs, something that a scalar reward model is inherently unable to offer. This advantage is reflected in **the consistently superior performance of our critique model under the critique-based refinement setting**.
>
> Table 2. Results of critique-based refinement against state-of-the-art ORM/PRM. "w2c" denotes the proportion of cases where a wrong
> solution becomes correct after judgment and refinement, while "c2w" indicates the opposite.
>
>
> |  | | Qwen2.5-7B-Instruct| ||||| Qwen2.5-72B-Instruct | ||||
> |------|:----:|:----:|:------:|:------:|:----:|:------:|:------:|:----:|:------:|:-----:|:----:|:------:|
> | || MATH500  | || AIME2425  | | |MATH500  | | | AIME2425 | |
> | | w2c | c2w | Acc. | w2c| c2w | Acc. | w2c| c2w | Acc. | w2c| c2w | Acc. |
> |***Before Refinement*** |
> ||-|-| 74.00| -|-|6.67 | - | - | 77.00 | -| -| 11.67|
> |***After Refinement*** |
> | Qwen2.5-Math-7B-PRM    | 1.20 | 0.40 | 74.80 | 3.33  | 0.00 | 10.00 | 3.80 | 1.00 | 79.80| 3.33| 1.67| 13.33     |
> |  Qwen2.5-Math-RM-72B  | 1.80 |0.20|75.60 | 3.33| 0.00| 10.00| 4.20| 0.80 | 80.40|3.33| 1.67 |13.33|
> |   DeepCritic-7B-RL | 5.40 | 2.00|  **77.40** |8.33| 1.67 |**13.33** | 6.00 |2.40| **80.60** |5.00 |1.67| **15.00**|
>
>
>
> Table 3. Results of verified majority voting over 32 samplings against state-of-the-art ORM/PRM (Verified Maj@32).
>
> |  | Qwen2.5-7B-Instruct|  | Qwen2.5-72B-Instruct | |
> |--------------|-----------:|-------------:| ------:| -----:|
> | | MATH500 |AIME24-25 | MATH500 | AIME24-25 |
> | None    |  83.80 | 15.00| 86.80 | 20.00 |
> | Qwen2.5-Math-7B-PRM   |85.80 |   **23.33**   |   **88.60**    | 20.00 |
> | Qwen2.5-Math-RM-72B  | 85.00  |    18.33  | **88.60** | **23.33** |
> |   DeepCritic-7B-RL | **86.00** | 20.00 | 88.00 | 20.00 |

---

> ### Author Response · Authors · 2025-11-21
> **Part 2**
>
> **Q3:** Regarding the quantitative analysis of deliberate critique behaviors during inference.
>
> **A3:** Following your suggestion, we analyze the proportion of self-correction behaviors exhibited in the critiques generated by our models across test sets of varying difficulty. Specifically, we first randomly sample 10 correct and 10 incorrect solutions from each of the four subsets of ProcessBench (resulting in ~100 reasoning steps for each set), together with the corresponding critiques generated by DeepCritic-7B-SFT and DeepCritic-7B-RL. We then prompte GPT-4o to determine, for each step-level critique, whether self-correction behavior is present. Finally, for each model and each subset, we calculate the proportion of step-level critiques that contained self-correction behaviors. The results are shown in the table below.
>
> Table 4. The statistics of self-correction behaviours of our models in inference time.
>
> | Subset | DeepCritic-7B-SFT | DeepCritic-7B-RL |
> |--------|------------------:|-----------------:|
> | GSM8K  |     23.4%     |          17.5%     |
> | MATH   |      27.6%          |      19.5%           |
> | OlympiadBench   |     24.6%     |      15.2%       |
> | OmniMath |   34.9%     |       17.0%          |
>
>
> As we can see, **both critique models indeed exhibit certain deliberate reasoning behaviours during inference**. Compared with the SFT model, the RL model exhibits a lower frequency of self-correction. A plausible explanation is that RL directly optimizes for judgment accuracy. As the model’s critique ability improves through RL training, its initial critiques become more accurate, thereby reducing the frequency for subsequent self-correction.
>
>
> -------
>
> **Q4:** Regarding the sensitivity of RL performance to alternative reward designs beyond binary accuracy (e.g., informativeness-aware rewards).
>
> **A4:** Thank you for your helpful question. Following your suggestion, we conduct additional experiments by augmenting the original judgment correctness reward with an additional informativeness-aware reward produced by an LLM judge. Specifically, during RL, we use ```Qwen3-4B-Instruct-2507``` to evaluate the content and the depth of each critique: (1) If a critique exhibits self-correction behavior that leads to the correct final judgment, we add +1 to the original accuracy reward. (2) if no self-correction behavior is present, the additional reward remains 0. (3) If self-correction occurs but leads to an incorrect final judgment, we add –1 to the original accuracy reward. The comparison results are shown below. We have two observations: (1) **Introducing informativeness-aware reward does not lead to improvements in judging performance.** Accuracy reward is the most relevant and effective signal for improving critique performance. (2) **Introducing an informativeness-aware reward substantially increases the average training response length from 2,000 to 2,800, which may indicate a form of *reward hacking***: the policy model produces excessive or uninformative self-reflection in an attempt to obtain higher informativeness-aware rewards, and this ineffective self-reflection directly contributes to the inflated response length.
>
> Table 5. The results of introducing informativeness-aware reward. "PB" refers to ProcessBench.
>
> |  | MR-GSM8K| PRM800K| PB-GSM8K| PB-MATH | PB-OlympiadBench | PB-OmniMath | Avg. |
> |--------|:----------:|:------:|:------:|:------:|:----:|:------:|:------:|
> | DeepCritic-7B-SFT  | 67.1 | 48.0 |   59.2 | 61.2  | 46.0 | 43.0 |  54.1 |
> | DeepCritic-7B-RL-Numina  |  **78.6** |  **57.1** | **75.2** | **70.0** | **54.3** | **51.2** |**64.4** |
> | DeepCritic-7B-RL-Numina (+ informativeness-aware reward)  |   76.9  | 51.2 |  67.6 |  62.4    | 49.7 | 46.3  | 59.0 |
>
> --------
>
> **Q5:** Regarding the generalization of weak-to-strong supervision when scaling to stronger generators (e.g., GPT-4o).
>
> **A5:** We follow your suggestion to use GPT-4o and GPT-4.1 as the generators, and evaluate the effectiveness of critique-based refinement by DeepCritic-7B-RL on supervising the stronger generators. The results are in the following table. We can see that **our critique model can also effectively supervise the outputs produced by stronger generators**.
>
> Table 6.  Results of critique-based refinement by using GPT-4o and GPT-4.1 as the generators. The results are evaluated on MATH500.
>
>
> | || GPT-4o  | | | GPT-4.1  | |
> |------|----:|----:|------:|------:|----:|------:|
> | | w->c| c->w | Acc. | w->c| c->w | Acc. | w->c| c->w | Acc. | w->c| c->w | Acc. |
> |***Before Refinement*** |
> ||-|-| 77.40 | - | - | 90.80 |
> |***After Refinement*** |
> | Qwen2.5-7B-Instruct   | 1.60   | 0.60 | 78.40  | 0.40 | 0.20 | 91.00 |
> |   DeepCritic-7B-RL | 3.60 |  1.20 | **79.80**  | 1.00 | 0.40 | **91.40** |

---

> > ### Author Response · Authors · 2025-11-21
> > **Part 3**
> >
> > **Q6:** Regarding the discussions on related critique enhancement methods.
> >
> > **A6:** Thank you for your question. Here, we provide a detailed discussion on the distinctions between our method and existing critique-enhancement approaches [1,2,3,4,5].
> >
> > First, the key distinction and advantage of our method over all previous critique-enhancement approaches lie in our **innovative in-depth critique generation procedure** introduced at the SFT data-curation stage, which is specifically designed to **encourage LLMs to develop more comprehensive and diverse critique behaviors**.
> > Prior methods [1,2,3,4] are essentially equivalent to performing SFT directly on initial critiques, which restricts the trained model to producing only direct verification–style critiques.
> > In contrast, our novel iterative critique generation process injects multi-perspective evaluation and meta-critiquing behaviours into the seed critique data.
> > This enables the trained model to perform deeper reasoning and to correct potential misjudgments that appear in earlier critique attempts. Thus， **our method enables substantially deeper critique depth**.
> >
> > Second, unlike some prior approaches [2,3,4] that rely sorely and heavily on large-scale SFT data to enhance the model’s critique capability, our method requires only a small amount of SFT supervision. Instead, we fully leverage the subsequent RL process, **allowing the model to autonomously explore and thereby unlock its own potential**.
> > In contrast to methods that depend on a powerful teacher model to generate massive quantities of critique data, our automated data-construction pipeline removes this dependency and is inherently **more scalable in terms of data production**.
> >
> > Third, we point out our method **is orthogonal to existing RL-based approaches [1,6] for improving LLM critique ability, and can therefore be complementary to them.**
> > For example, works such as [1,6] reinforce an LLM critic by using the refinement accuracy of an LLM generator—conditioned on the critic's critiques—as the reward signal. In contrast, we adopt a fundamentally different problem formulation. By automatically annotating the first incorrect step for each solution, we transform the problem into a standard generative reward modeling setting. This allows us to train a critique model that is both efficient and reliable. Importantly, the critique model produced by our approach can serve as a strong initialization for the frameworks in [1,6], alleviating the common issue where the initial critic can generate only shallow critiques. Building on top of our model, their RL pipelines can further enhance performance, making the two directions naturally complementary.
> >
> > Finally, the work Critique-guided Distillation [5] does not fall into the category of enhancing an LLM’s critique ability. Instead, it directly leverages an existing LLM critic to guide and improve the training of LLM generators.
> >
> > We will revise the corresponding part in the related work to highlight the unique advantages of our work.
> >
> > -----
> >
> >
> > **We hope the above response addresses your questions. We are glad to continue the discussion if you have any further questions. We are incorporating above new results into the revision, and will update the submission once the revision is ready.**
> >
> >
> > [1] Xie, Zhihui, et al. "Teaching language models to critique via reinforcement learning." ICML 2025
> >
> > [2] Wei, Jiaqi, et al. "Alignrag: An adaptable framework for resolving misalignments in retrieval-aware reasoning of rag." arxiv 2025
> >
> > [3] Wang, Yubo, Xiang Yue, and Wenhu Chen. "Critique fine-tuning: Learning to critique is more effective than learning to imitate." arxiv 2025
> >
> > [4] Wang, Yubo, et al. "Unleashing the Reasoning Potential of Pre-trained LLMs by Critique Fine-Tuning on One Problem." EMNLP 2025
> >
> >
> > [5] Kapusuzoglu, Berkcan, et al. "Critique-Guided Distillation: Improving Supervised Fine-tuning via Better Distillation." arxiv 2025
> >
> > [6] Xi, Zhiheng, et al. "Critique-RL: Training Language Models for Critiquing through Two-Stage Reinforcement Learning." arxiv 2025

---

> > > ### Author Response · Authors · 2025-11-27
> > > **Looking forward to your feedback**
> > >
> > > Dear Reviewer Err4,
> > >
> > > We sincerely thank you for reviewing our paper and providing helpful comments. We have addressed your questions in our rebuttal. We would like to know whether if you have any further questions, and we are glad to continue the discussion if there are any.
> > >
> > > We are sincerely looking forward to your feedback, and your support means a lot to us! Thank you.
> > >
> > > Best regards,
> > >
> > > Authors

---

### Author Response · Authors · 2025-12-03
**Rebuttal and Revision Summary**

Dear Area Chairs,

We sincerely thank you for your additional efforts on reviewing our submission and making the decision. During the rebuttal phase, we addressed all the questions from each reviewer and uploaded the revised version of our manuscript, with the changes highlighted in blue. To help you better and more efficiently handle our submission and the rebuttal process, we are writing this message to provide a summary of the reviews and outline how we have addressed each of the raised concerns.

---

### Summary of our paper and reviews


Our paper proposes **a novel and effective framework that enables LLMs to perform deliberate critiques for model outputs, paving the way for achieving scalable oversight**. We are encouraged by the following recognitions from the reviewers:

- **Important and timely research problem.** (Reviewer ```y2i7```)

- **Novel and insightful methodology.** (Reviewer ```Err4``` and ```y2i7```)

- **Comprehensive and strong experiments to demonstrate the great effectiveness and generalizability of the method.** (all reviewers)

---

### Summary of rebuttal and revision

During the rebuttal phase, we made sustained efforts to address all the reviewers' questions and have uploaded the revised version, which includes new results and analysis.  Below, we first summarize our responses to some common questions raised by the reviewers.

> **Q1: Regarding the discussion on the unique advantages of our work compared to previous critique-enhancement approaches.** (Reviewer ```Err4``` and ```PU76```)

A1: We point out that the key distinction and advantage of our method over all previous critique-enhancement approaches lie in our **innovative in-depth critique generation procedure** introduced at the SFT data-curation stage, which is specifically designed to encourage LLMs to develop more comprehensive and diverse critique behaviors. We have included the detailed discussion in the Related Work in the revision.

>  **Q2: Regarding the concern on the the dependency on a strong teacher model for seed data generation.** (Reviewer ```PU76``` and ```y2i7```)

A2: We would like to point out that our method can be applied for self-improvement of LLMs without requiring stronger teacher models. We have included the self-improvement experiments in Section 4.4 to validate this.

> **Q3: Regarding the generalization of our approach to subjective domains.** (Reviewer ```PU76``` and ```y2i7```)

A3: We have followed the reviewers' questions to conduct additional experiments in a subjective domain, *text summarization*, to demonstrate the effectiveness of generalizability of our method to these open domains beyond verifiable tasks. The corresponding results and analysis are in Appendix M.

> **Other key responses to individual questions:**

- We show the statistics of problem sources in the auto-constructed RL data in Table 7 to demonstrate the diverse distribution of problem difficulty in our final dataset. (Reviewer ```Err4```)

- We display the comparison results against advanced ORM/PRMs to strengthen the effectiveness of our method. (Reviewer ```Err4```)

- We display the quantitative analysis of deliberate critique behaviors of our models during inference in Figure 3 to validate the effectiveness of our meta-critiquing practice. (Reviewer ```Err4```)

- We show the ablation results of introducing informativeness-aware reward during RL in Appendix L.  (Reviewer ```Err4```)

- We show the results of critique-based refinement on GOT-4o and GPT-4.1 to show that our critique model can also effectively supervise the outputs produced by stronger generators. (Reviewer ```Err4```)

- We show the ablation results of performing RL on noisy data in Appendix K. (Reviewer ```PU76```)

- We show the ablation results of using binary search for label auto-annotation in Appendix J to further strengthen the scalability of our framework. (Reviewer ```PU76```)

----

We sincerely thank all the reviewers for their insightful comments and helpful suggestions. We also thank you once again for your additional efforts in the review process. We hope the above summary can assist you in reviewing our submission more efficiently.


Best regards,

Authors of Submission 11560

---

### Meta-Review · Area_Chair_9duX · 2026-01-07

**Summary:**

While the empirical results are strong, the underlying framework adopted in this paper is well-established in the reasoning domain. As noted by Reviewer PU76, this work can be viewed primarily as an application of existing paradigms to the critique capability. Furthermore, the proposed data curation pipeline entails prohibitive computational costs that hinder scalability, a concern that was not convincingly justified in the rebuttal.

**Reviewer Concerns:**

1. Addressed Concerns:
* Generalization: In response to Reviewers PU76 and y2i7 regarding the focus solely on the mathematical domain, the authors added experiments on text summarization (Subjective Domain) in Appendix M, preliminarily demonstrating the method's cross-domain potential.
* Missing Baselines: Addressing Reviewer Err4's suggestion, the authors added comparisons with advanced Outcome Reward Models (ORMs) and hybrid PRM-ORM setups, strengthening the empirical evidence.
* Teacher Dependency: The authors responded to Reviewer PU76's concern about reliance on the 72B teacher model by adding "Self-Improvement" experiments (generating data using the 7B model itself).
2. Outstanding Concerns:
* Limited Novelty (Reviewer PU76): Although the authors emphasized the innovation of the "In-depth Meta-Critique" data construction process, Reviewer PU76 pointed out that fundamentally, the framework remains an application of a mature paradigm to a specific task. It lacks significant breakthrough innovation at the algorithmic or architectural level, a point that stands even after the rebuttal.
* Cost & Scalability (Reviewer y2i7): The method relies on Monte Carlo sampling at every step of the reasoning chain to locate errors. This linear scanning approach incurs extremely high computational costs. Reviewer y2i7 explicitly identified this as a major scalability bottleneck, which remains unresolved.

**Reviewer Scores:**

Many of Reviewer Err4’s specific technical questions were largely addressed in the authors’ responses, so if the reviewer had participated actively in the discussion, they would likely have increased the score slightly.

---

### Decision · Program_Chairs · 2026-01-26

Reject